

# A Singular Value Decomposition framework for retrievals with vertical distribution information from greenhouse gas column absorption spectroscopy measurements

Anand K. Ramanathan[1,2], Hai M. Nguyen[3], Xiaoli Sun[2], Jianping Mao[1,2], James B. Abshire[2], Jonathan M. Hobbs[3], and Amy J. Braverman[3]

[1]Earth System Science Interdisciplinary Center, University of Maryland, College Park, MD - 20740
[2]NASA Goddard Space Flight Center, 8800 Greenbelt Road, Greenbelt, MD - 20771
[3]NASA Jet Propulsion Laboratory, California Institute of Technology, 4800 Oak Grove Drive, Pasadena, CA - 91109

*Correspondence to:* Hai Nguyen (hai.nguyen@jpl.nasa.gov)

**Abstract.** We review the singular value decomposition (SVD) framework and use it for quantifying and discerning vertical information in greenhouse gas retrievals from column integrated absorption measurements. While the commonly used traditional Bayesian optimal estimation (OE) assumes a prior distribution in order to regularize the inversion problem, the SVD approach identifies principal components that can be retrieved from the measurement without explicitly specifying a prior mean and prior covariance matrix. We review the SVD method, explicitly recognize the use of an uninformative prior and show it to be bias-free in the absence of forward model error irrespective of the choice of the uninformative prior. We also make the connection between the SVD method and the pseudo-inverse, which makes it more intuitive and easy to understand. We illustrate the use of the SVD method on an integrated path differential absorption $CO_2$ lidar measurement model, and verify our derivations and bias free properties versus optimal estimation using numerical simulations. In contrast, traditional OE retrievals exhibit bias when the prior mean used in the retrieval differs from the true mean. Hence, the SVD method is particularly useful for situations where knowledge of the prior mean and prior covariance of the true state (e.g., greenhouse gas profiles) is inadequate.

## 1 Introduction

In the past few decades, anthropogenic climate change has brought a renewed interest in carbon cycle science, and thus for accurate sensing of greenhouse gas (GHG). GHG column remote sensing measurements are made using satellite-based optical spectrometers such as those aboard the Greenhouse gas Observing Satellite (GOSAT, Kuze et al. (2009)) and the Orbiting Carbon Observatory (OCO-2, Boesch et al. (2011)), ground-based spectrometers such as the Total Column Carbon Observing Network (TCCON, Wunch et al. (2011) and other instruments (Gisi et al., 2012). Atmospheric measurements have also been made using airborne integrated path differential absorption (IPDA) lidar instruments (Abshire et al., 2014; Lin et al., 2015; Menzies et al., 2014; Refaat et al., 2016). While column-averaged mixing ratios are retrieved from measurements using methods ranging from simple differential absorption ratioing (Refaat et al., 2016), least-squares line-fitting (Wunch et al., 2011) and





traditional optimal estimation (OE) (Connor et al., 2008), information about the GHG vertical distribution (which we shall refer to as vertical information) is more difficult to obtain and typically not routinely reported as part of GHG retrievals.

Although in principle, the traditional OE (Rodgers, 2000) is capable of extracting vertical information in the measurement, in practice the assumption of a prior GHG distribution, which is necessary for the regularization of the problem, makes the

retrieval potentially bias-prone. Here, by traditional OE we mean an application of the optimal retrieval framework as described in Rodgers (2000) where the input prior covariance matrix is informative (i.e., the prior covariance matrix has at least one finite eigenvalue). In contrast, the Singular Value Decomposition (SVD) approach Hansen (1990) can extract vertical information from the measurement without assuming any prior GHG distribution, opening the possibility of an unbiased retrieval. The SVD method is based on retrieving the leading principal components of the trace gas mixing ratio state vector from the

measurement. The vertical information contained in the principal components can provide useful information for carbon flux inferences, thanks to the correlations between the pressure broadening (and thus absorption lineshape) of two layers and their GHG mixing ratios (due to GHG vertical transport).

The theoretical basis of the SVD method has been previously laid out in the context of the general underdetermined inversion problem (Hansen, 1990). Rodgers (2000) also has a discussion on the topic. Borsdorff et al. (2014) present a review of the SVD

and related methods in the context of trace gas retrievals and the connections to the traditional OE as well as simple profile scaling methods. The SVD method has also been applied to remote sensing for ozone (Hasekamp and Landgraf, 2001) and methane (Butz et al., 2010). Previous work has used the SVD method primarily to regularize the underdetermined retrieval problem, but also for computational efficiency and to eliminate the need for knowledge of the prior distribution.

In this work, we choose a specific greenhouse gas measurement system and study the principal components and illustrate

how they provide useful, quantifiable information about the vertical distribution of the gas. In choosing to evaluate the retrieval method via the principal components, the implicit prior used is strictly uninformative and the retrieved principal components are thus bias-free, which we explicitly show. In addition, we explore the instrument spectral resolution necessary to obtain vertical information. Finally, we illustrate the theory using numerical simulations.

This paper also attempts to make the theoretical framework of the SVD method more accessible to readers who may not be

as familiar with the matrix algebra conventions used in books like Rodgers (2000). It should be noted that many of the articles cited in Table 1 use non-matrix equations for performing retrievals, even though the matrix formalism is more complete and general. By choosing a relatively simple $CO_2$ IPDA lidar system to focus on, we are able to make a direct connection between the retrieval problem and the underlying physics, with no major assumptions or simplifications. We also illustrate the most important matrices so that the reader is able to get an intuitive sense of the physics beneath the matrix algebra.

The SVD method works similar to least-squares line-fitting retrieval approaches but offers a more formal framework (Borsdorff et al., 2014). Here, we extend the approach to retrieve vertical GHG profile information without incurring bias from the regularization process. In contrast, the regularization process in the traditional OE method incurs bias when the prior GHG vertical profile is not close to the true GHG vertical profile. Biases are a concern for atmospheric carbon dioxide ($CO_2$) measurements, since even small biases are known to affect carbon flux inversions (Chevallier et al., 2014).





**Table 1.** Comparison of retrieval algorithms used for GHG remote sensing based on regularization method and source of vertical information. The approximate spectral resolution (instrument linewidth, see section 4.4), is given in brackets for each type of measurements. The SVD method proposed in this work extracts information of the vertical GHG distribution strictly from the measurement, making no assumption of a prior distribution. Note that use of a uniform column for vertical information is equivalent to the use of an uninformative prior. * - Butz et al. (2011) used the prior distribution for regularizing the $CO_2$ retrievals but the SVD-reduced dimensionality for $CH_4$

| Measurement | Instrument | Reference (Algorithm) | Regularization method | Column Average | Profile info. |
|---|---|---|---|---|---|
| GHG Satellites ($\sim 10^{-1}$cm$^{-1}$) | SCIAMACHY ($CO_2$) | Reuter et al. (2010) ($CO_2$), Frankenberg et al. (2006) ($CH_4$) | Prior distr. | prior+meas | prior+meas |
| | GOSAT ($CO_2$, $CH_4$) | Kuze et al. (2009) (NIES), Crisp et al. (2012) (ACOS), Butz et al. (2011) (RemoTeC) | Prior distr.* | prior+meas | prior+meas |
| | OCO-2 ($CO_2$) | Connor et al. (2008) | Prior distr. | prior+meas | prior+meas |
| Ground-based Spectrometers, ($\sim 10^{-2}$cm$^{-1}$) | TCCON ($CO_2$, $CH_4$, CO, $N_2O$) | Wunch et al. (2011) (GFIT) | Fixed profile | measurement | prior |
| | TCCON ($CO_2$) | Kuai et al. (2012); Dohe (2013) | Prior + Reduced Levels | prior+meas | prior+meas |
| | TCCON ($CO_2$) | Connor et al. (2016) (GFIT2) | Prior distr. | prior+meas | prior+meas |
| | TCCON ($CH_4$) | Tukiainen et al. (2016) | SVD-Reduced Dim | prior+meas | prior+meas |
| | Bruker EM27 ($CO_2$) | Gisi et al. (2012) (GFIT) | Fixed profile | measurement | prior |
| | Mini-LHR ($CO_2$) | Melroy et al. (2015) | Fixed profile | measurement | uniform |
| | Others spectrometers ($CO_2$, $CH_4$) | Yuan et al. (2015); Si-Yang et al. (2013); Petri et al. (2012) | Fixed profile | measurement | prior |
| Airborne IPDA lidars , ($< 10^{-3}$cm$^{-1}$) | $CO_2$ MFLL | Lin et al. (2015) | Fixed profile | measurement | prior |
| | $CO_2$ LAS | Menzies et al. (2014) | Fixed profile | measurement | prior |
| | $CO_2$ Sounder | Abshire et al. (2014) | Fixed profile | measurement | uniform |
| | $CH_4$ Sounder | Riris et al. (2012) | Fixed profile | measurement | uniform |
| | 2-$\mu$m $CO_2$ IPDA | Refaat et al. (2016) | Fixed profile | measurement | prior |
| | CHARM-F | Amediek et al. (2017) | Fixed profile | measurement | prior |
| This work | space lidar model | | SVD-Reduced Dim | measurement | measurement |

The paper is organized as follows. In section 2, we introduce the problem of regularization, which is intimately tied in to the challenge of extracting information about the vertical distribution, and set up the radiative transfer equations and retrieval equations. We follow it up in section 3 with a description of the SVD method, its ability to extract vertical information and its robustness against bias in the absence of prior information on the GHG vertical profile. In section 4, we apply the SVD method to the specific case of the $CO_2$ Sounder lidar instrument, and proceed in section 5 to perform numerical simulations comparing the SVD and traditional OE methods. We then describe the implications of this work in section 6 before concluding.



## 2 Retrievals from GHG absorption measurements

A retrieval seeks to extract certain information from a measurement. Retrieval problems may or may not be fully determined. In situations where the retrieval problem is fully determined, one can perform a least-squares fit to solve for the parameters of interest. However, for column GHG absorption measurement spectra obtained from remote sensing, the retrieval is generally

underdetermined, and thus needs some kind of regularization to make it more deterministic.

### 2.1 Regularization of the retrieval problem and vertical information

The traditional Bayesian OE method (Rodgers, 2000) recommends linearization of the problem close to the solution followed by regularization by a term corresponding to a prior distribution for the state. SVD and related methods (Hasekamp and Landgraf, 2001) perform an unconstrained retrieval, equivalent to the use of an uninformative prior, on subspaces of the trace

gas column that are informed by the measurement. These regularization methods allow a solution to be computed, but may also induce bias on either certain dimensions (SVD and related methods) or all dimensions (traditional OE) of the solution space.

At this point it is useful to qualify what we mean by prior information. The use of prior information in some form is unavoidable in any kind of GHG remote sensing retrieval, since it is not possible to simultaneously measure all the parameters needed for determining the GHG mixing ratio. For instance, the absorption depends on the spectroscopic parameters, which are

determined from laboratory measurements, and the atmospheric pressure and temperature profile, which are typically obtained from weather models. A comprehensive quantification of uncertainty that includes errors arising from all these sources of "prior" information is well beyond the scope of this work. Rather, we will focus on how the assumption of a prior distribution affects the retrieved estimate of the GHG profile.

Although traditional OE has become the de facto standard for satellite GHG remote sensing (Oshchepkov et al., 2013),

ground based spectrometers and airborne IPDA lidar (see Table 1) have largely avoided it and other regularization methods by resorting to dimension reduction. Typically, a fixed profile shape is assumed (Wunch et al., 2011; Abshire et al., 2014), and only a simple vertical profile scaling parameter is retrieved. Such simple methods have the advantage of better representing the instrument measurement, and enabling more feedback on instrument performance. Despite preliminary evidence to the contrary (Wunch et al., 2010), there remains the open question of whether biases are introduced by the assumption of a fixed vertical

GHG profile, the potential under-fitting of the absorption spectrum and the failure to exploit all the information contained in the measurement. In addition, this simple scaling of a vertical profile also precludes such instruments from discerning any information about the vertical GHG distribution.

Between the traditional OE retrieval and the least-squares fitting via simple scaling method, there exist some intermediate choices. In a recent advance, Kulawik et al. (2016) extract the GHG mixing ratio of two vertical layers from GOSAT data using

the OE method with a reduced vertical basis and an uninformative prior. The authors choose to use an uninformative prior for regularization to ensure that any vertical information can be attributed to the measurement alone. There have also been attempts to retrieve vertical information from ground-based sun spectrometer measurements by easing the constraints imposed by OE, as given in Wunch et al. (2011). Kuai et al. (2012) and Dohe (2013) used a reduced number of vertical levels and applied additional





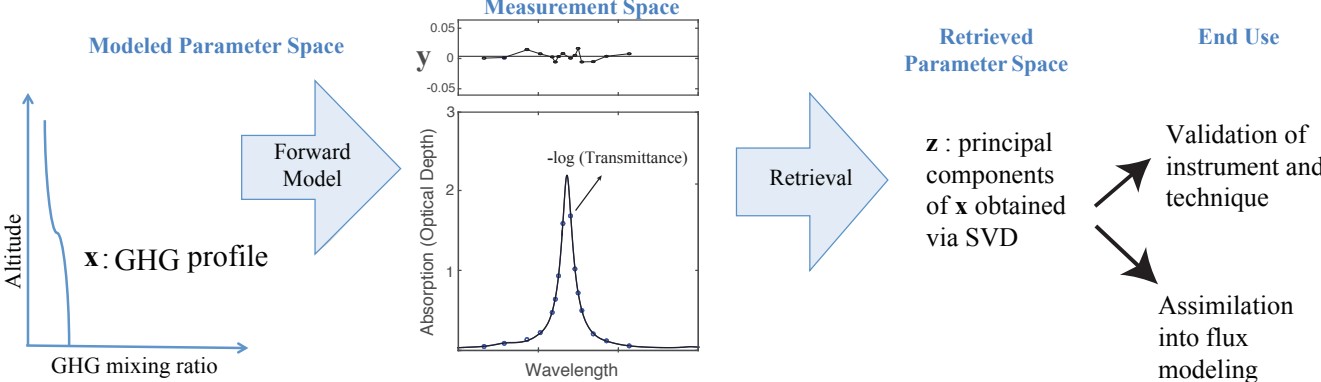

**Figure 1.** (Color online) Schematic of the various terms involved in a greenhouse gas (GHG) measurement, retrieval and end use. The Singular Value Decomposition (SVD) method introduces a new retrieval basis space **z**, which is different from the model parameter space **x**. In using the **z** basis, the SVD retrieval makes no assumptions regarding the prior GHG distribution, thus avoiding a potential source bias and making the validation and flux modeling more straightforward.

constraints via the choice of the prior covariance matrix. In fact, Cressie et al. (2017) show that, for a fully determined problem, the (non-Bayesian) least-squares fit is simply a special case of the optimal estimate using an uninformative prior. Thus, one can move back and forth along the spectrum of retrieval methods from fully-Bayesian to non-Bayesian by combining the choice of the prior covariance matrix and the choice of dimension of the basis describing vertical structure.

Dimension reduction via SVD has been previously used both for satellite retrievals (Masiello et al., 2012; Thompson, 1992; Butz et al., 2010), ground-based spectrometers (Tukiainen et al., 2016) and laboratory laser absorption measurements (Bomse and Kane, 2006). The SVD approach described here comes closest to the one applied for satellite methane retrievals (Butz et al., 2010), but performs the retrieval in the principal component basis to eliminate bias originating from the choice of the uninformative prior used (see section 3.5). Components in the reduced dimensional principal component space can be directly

assimilated into flux models similar to the way $X_{CO2}$ is presently assimilated (Basu et al., 2013).

## 2.2 The Radiative Transfer Problem

Remote sensing measurements of GHGs are typically assimilated into a carbon flux inversion system or other modeling (see Figure 1). We set up the radiative transfer problem and retrieval keeping in mind that the measurements are not an end in themselves. In addition, to best illustrate the SVD method, we choose a simplified measurement geometry and atmospheric

conditions, all of which are satisfied by a nadir pointed IPDA lidar instrument such as Abshire et al. (2014) :

1. Nadir sounding geometry with light traveling along a perfect vertical path - Lidar instruments satisfy this condition since they are pointed nadir and have the source and detector on the same platform.



2. Perfect knowledge of the optical path length with a clear atmosphere - Lidar instruments are pulsed (Abshire et al., 2014; Refaat et al., 2016), or alternatively have some modulation (Lin et al., 2015)), and simultaneously measure the surface elevation (via ranging) and thus the precise light path length. In addition, this ranging capability enables the time-gating of the surface returns so as to exclude aerosol backscatter, a common cause of bias.

3. Undistorted measure of atmospheric transmittance with negligible instrument broadening - Lidar instruments have a narrow laser linewidths, which determines their instrument lineshape function, which typically 3-4 orders of magnitude narrower than spectrometers. The laser line width is negligible compared to the molecular absorption lineshape and can be assumed monochromatic.

4. Negligible interference from other atmospheric species via careful line choice - Lidar instruments typically sample a
single absorption line, rather than a full absorption band. For this narrow spectral range, absorption from other species can be ignored.

5. Sufficient number of wavelength samples - Due to complexities in generating precisely tuned laser light for wavelength samples, many lidar GHG sensing instruments (Refaat et al., 2016; Lin et al., 2015; Menzies et al., 2014) use only 2 wavelength samples. Here, we assume at least a few wavelength samples across the absorption line.

**We divide the atmosphere into** $m$ **layers**. We make the layers in equal intervals of pressure to keep the number of air molecules in each layer the same. The atmospheric transmittance can be expressed as the negative exponent of the sum of the absorption (expressed in optical depth units) of the individual layers of height $h_i$:

$$T(\lambda, \mathbf{x}, \mathbf{b}, h) = \exp\left(-\sum_{i=1}^{m} x_i OD(\lambda, \mathbf{b_i}) h_i\right), \tag{1}$$

where $T$ is the two-way transmittance, $OD(\lambda, \mathbf{b_i})$ represents the spectroscopic model calculating the two-way GHG absorption
in units of optical depth per distance at wavelength $\lambda$ for the atmospheric conditions $\mathbf{b_i}$ (consisting of the atmospheric pressure and temperature). $\mathbf{b}$ is a vector containing the profiles $\mathbf{b_i}$. $\mathbf{x}$ is the vector containing the GHG mixing ratio profile $x_i$. The total path length $h = \sum h_i$ is given in units of distance.

Next we define a measurement vector $\mathbf{y}$ **consisting of** $n$ **samples of an absorption line**, and define the measurement equation with noise assuming perfect knowledge of the forward model,

$$\mathbf{y} = \mathbf{F}(\mathbf{x}) + \epsilon, \tag{2}$$

and the forward model,

$$\mathbf{F}(\mathbf{x}) = \begin{bmatrix} x_0 - \log\left(\frac{T(\lambda_1, \mathbf{x})}{T(\lambda_1, \mathbf{x}_u)}\right) \\ \vdots \\ x_0 - \log\left(\frac{T(\lambda_n, \mathbf{x})}{T(\lambda_n, \mathbf{x}_u)}\right) \end{bmatrix}. \tag{3}$$





$\epsilon$ represents the measurement noise, which will be described in section 2.4. The atmospheric conditions and absorption path have been assumed fixed for each sounding and thus left out of the explicit notation. We have incorporated a measurement amplitude $x_0$ term, which includes all signal attenuation and loss factors, in the vector $\mathbf{x}$.

Additionally, we have normalized $T(\lambda, \mathbf{x}, z)$ by $T(\lambda, \mathbf{x}_u, z)$, where $\mathbf{x}_u$ is the uninformative prior. We have also taken the natural logarithm to make the problem linear with respect to the change in the GHG concentration $\mathbf{x}$, enabling the use of the tools of linear algebra. With that, $\mathbf{y}$ is defined as the *deviation* in the absorption from that of a column defined by $\mathbf{x}_u$ rather than the absorption itself. A schematic of the model parameter and measurement spaces,

$$\mathbf{x} = \begin{bmatrix} x_0 \\ x_1 \\ \vdots \\ x_m \end{bmatrix} \quad \text{and} \quad \mathbf{y} = \begin{bmatrix} y_1 \\ \vdots \\ y_n \end{bmatrix}$$

respectively, is given in Figure 1.

As with most atmospheric measurements, the retrieval problem for GHG remote sensing cannot be expressed as a non-singular analytic expression based on the forward model. In the remainder of this section, we will set up the retrieval problem analogous to Rodgers (2000) and define the various matrices needed for the solution.

### 2.3 Forward model Jacobian

Having set up the radiative transfer problem in Eq. (1) and defined the forward model in Eq. (3), one can see that the problem
is already linear with respect to the change in GHG concentration. For problems that are not linear, one can now take the linear approximation for small perturbations, a standard technique used extensively by Rodgers (2000).

We use an uninformative prior, $\mathbf{x_0}$. As we will later show mathematically (section 3.5) and through numerical simulations (section 5), the retrieval in the principal component basis is insensitive to the choice of the uninformative prior. We can now express the measurement vector as

$$
\begin{aligned}
\quad \mathbf{y} &= \mathbf{F}(\mathbf{x}_u) + \frac{\partial \mathbf{F}(\mathbf{x})}{\partial \mathbf{x}}(\mathbf{x} - \mathbf{x}_u) \\
&= \mathbf{F}(\mathbf{x}_u) + \mathbf{K}(\mathbf{x} - \mathbf{x}_u),
\end{aligned}
\tag{4}
$$

$$
\mathbf{K} = \begin{bmatrix}
1 & OD(\lambda_1, \mathbf{b_1})h_1 & \ldots & OD(\lambda_1, \mathbf{b_m})h_m \\
1 & OD(\lambda_2, \mathbf{b_1})h_1 & \ldots & OD(\lambda_2, \mathbf{b_m})h_m \\
\vdots & \vdots & \vdots & \vdots \\
& OD(\lambda_n, \mathbf{b_1})h_1 & \ldots & OD(\lambda_n, \mathbf{b_m})h_m
\end{bmatrix}
\tag{5}
$$

where $\mathbf{K}$ is an $n \times (m+1)$ matrix of partial derivatives. $\mathbf{K}$ is termed as the Jacobian or Kernel matrix.





### 2.4 Measurement Noise Matrix

The measurement $\mathbf{y}$ is associated with noise, which we characterize using the measurement error covariance matrix (Rodgers, 2000) $\mathbf{S}_\epsilon$, which has dimensions $n \times n$. The noise is assumed to be Gaussian (random noise only) and the diagonal elements of $\mathbf{S}_\epsilon$ represent the variance (in a large sample of identical, repeated observations) of the individual wavelength samples. For a

perfect instrument, which we assume here, the off-diagonal terms, which represent covariances between different wavelength samples are zero.

$$
\mathbf{S}_\epsilon = \begin{bmatrix} \langle(y_1 - \langle y_1\rangle)^2\rangle & 0 & \dots & 0 \\ 0 & \langle(y_2 - \langle y_2\rangle)^2\rangle & \dots & 0 \\ \vdots & \vdots & \dots & \vdots \\ 0 & 0 & \dots & \langle(y_n - \langle y_n\rangle)^2\rangle \end{bmatrix}
$$

### 2.5 Retrieval Equations

To derive an estimate of the state $\mathbf{x}$ from measured radiance $\mathbf{y}$, we define a loss function (or weighted least-squares error) as

follows,

$$
L(\mathbf{x}) = [\mathbf{y} - \mathbf{F}(\mathbf{x})]^T \mathbf{S}_\epsilon^{-1}[\mathbf{y} - \mathbf{F}(\mathbf{x})]. \tag{6}
$$

Note that Eq. (6) is the same as the method of least squares, except here we are weighting the sum of squared error by the measurement error matrix $\mathbf{S}_\epsilon$. This weighted sum of squared errors is widely used in regression frameworks, and it is the loss function of choice for retrievals that are not based on optimal estimation (e.g., Atmospheric Infra-Red Sounder or AIRS,

Chahine et al. (2005) and Cressie et al. (2017)). In contrast to the more common Bayesian treatments of the problem (Rodgers, 2000), we are not required to explicitly specify the *a priori* distribution for $\mathbf{x}$ in Eq. (6).

To find the optimum $\mathbf{X}$, we take the derivative of $L(\mathbf{x})$ with respect to $\mathbf{x}$:

$$
\begin{aligned}
\frac{dL}{d\mathbf{x}} &= 2\mathbf{K}^T\mathbf{S}_\epsilon^{-1}[\mathbf{y} - \mathbf{K}\mathbf{x}]. \\
&= \mathbf{K}^T\mathbf{S}_\epsilon^{-1}\mathbf{y} - \mathbf{K}^T\mathbf{S}_\epsilon^{-1}\mathbf{K}\mathbf{x}
\end{aligned}
$$

For simplicity, with no loss in generality, we have set

$$
\mathbf{x}_u = 0, \text{ and } \mathbf{F}(\mathbf{x}_u) = 0
$$

which imply that $\mathbf{x}$ now represents the *deviation* from the GHG profile and signal level in $\mathbf{x}_u$, and $\mathbf{y}$ the deviation from $\mathbf{F}(\mathbf{x}_u)$. The optimal state vector $\hat{\mathbf{x}}$ that minimizes the loss function in Eq. (6) can be found by setting the derivative to 0 and solving as follows,

$$
\begin{aligned}
\mathbf{K}^T\mathbf{S}_\epsilon^{-1}\mathbf{y} - \mathbf{K}^T\mathbf{S}_\epsilon^{-1}\mathbf{K}\hat{\mathbf{x}} &= 0 \\
\mathbf{K}^T\mathbf{S}_\epsilon^{-1}\mathbf{y} &= \mathbf{K}^T\mathbf{S}_\epsilon^{-1}\mathbf{K}\hat{\mathbf{x}}
\end{aligned} \tag{7}
$$





Equation 7 can be used to solve for the optimal estimate $\hat{\mathbf{x}}$ from a single measurement $\mathbf{y}$. Solving for a unique $\hat{\mathbf{x}}$ is usually not possible since it requires the inversion of the matrix $\mathbf{K}^T\mathbf{S}_\epsilon^{-1}\mathbf{K}$, which is typically singular. This implies that the complete information required to retrieve a unique $\hat{\mathbf{x}}$ is not present in the measurement $\mathbf{y}$. The standard practice, as described in Rodgers (2000) is to use *a priori* information to regularize Eq. (7), but here we will explore the alternative SVD method.

## 3 The Singular Value Decomposition approach

The Singular Value Decomposition (SVD) approach (Hansen, 1990) involves regularizing Eq. (7) by only solving for the principal components of the $(m+1)\times(m+1)$ matrix $\mathbf{K}^T\mathbf{S}_\epsilon^{-1}\mathbf{K}$. Alternatively, it can be interpreted as inverting Eq. (7) using a reduced-rank pseudo-inverse (discussed in section 3.3). Matrix SVD is a standard tool in matrix algebra whose applications include least-squares fitting, principal component analysis (Wall et al., 2003; Madsen et al., 2004) and calculating the pseudo-inverse of a matrix, all of which are related to the approach used here.

Before getting into the formal derivation of the principal component basis, it is useful to bring in some physical intuition. The nature of the principal components are tied to the lineshapes of the various atmospheric layers. Pressure broadening of the lineshape in the atmosphere leads to the first principal component being shaped like a "mean" lineshape, and representing a sort of column average. Higher order principal components represent higher order moments in the atmospheric profile, and as one would expect are more challenging to measure.

The remainder of this section formally reviews and describes the SVD framework along the lines of Butz et al. (2010). In contrast to previous SVD work (Hansen, 1990; Hasekamp and Landgraf, 2001; Butz et al., 2010), we describe the mathematics underlying the SVD approach using the retrieval basis $\mathbf{z}$ of the principal components of $\mathbf{K}^T\mathbf{S}_\epsilon^{-1}\mathbf{K}$, which we will refer to as the principal component basis. In section 3.3, we connect the SVD retrieval method to a rank-reduced pseudo-inverse applied to the retrieval equation. In section 3.5, we show how using the SVD method with the principal component basis can avoid bias from regularization and thus render the prior truly uninformative. Readers with a preference for an intuitive understanding based on the underlying physics can, as they read along, refer to section 4, which illustrates the SVD framework applied to a specific instrument and measurement.

### 3.1 The z Retrieval Basis of Principal Components

To calculate the principal component basis $\mathbf{z}$, we perform a singular value decomposition (Wall et al., 2003) of the matrix $\mathbf{S}_\epsilon^{-\frac{1}{2}}\mathbf{K}$:

$$\mathbf{S}_\epsilon^{-\frac{1}{2}}\mathbf{K} = \mathbf{U}\boldsymbol{\Gamma}\mathbf{V}^T, \tag{8}$$

where

- $\mathbf{U}$ is an $n \times n$ orthogonal matrix (rows consist of unit vectors that are normal to each other)

- $\boldsymbol{\Gamma}$ is an $n \times (m+1)$ matrix having all non-main diagonal elements $(i,j : i \neq j)$ equal to zero





  – $\mathbf{V}$ is an $(m+1) \times (m+1)$ orthogonal matrix

The matrix singular value decomposition described in Eq. (8) is a standard function available in most numerical software packages. It is also equivalent to extracting the principal components of $\mathbf{K}^T \mathbf{S}_\epsilon^{-1} \mathbf{K}$ via eigenvector decomposition. In a singular value decomposition, the first few rows of $\mathbf{V}^T$ capture the most significant information contained in $(\mathbf{S}_\epsilon^{-\frac{1}{2}} \mathbf{K})$, and thus by changing and reducing the basis of $\mathbf{x}$, we can obtain a unique solution to (7).

The new principal component $\mathbf{z}$ basis is defined by

$$
\begin{aligned}
\mathbf{z} &= \tilde{\mathbf{V}}^T \mathbf{x}, \quad \text{where} \\
\tilde{\mathbf{V}}^T &= \tilde{\mathbf{I}}_{m+1,p}^T \mathbf{V}^T
\end{aligned}
\tag{9}
$$

$$
\tilde{\mathbf{I}}_{m+1,p} =
\begin{bmatrix}
1 & 0 & \dots & 0 \\
0 & 1 & \dots & 0 \\
\vdots & \vdots & \ddots & 0 \\
0 & 0 & \dots & 1 \\
\vdots & \vdots & \vdots & \vdots \\
& 0 & \dots & 0
\end{bmatrix}
$$

where $\tilde{\mathbf{V}}^T$ is a row-truncated version of $\mathbf{V}^T$. Both $\tilde{\mathbf{V}}$ and $\tilde{\mathbf{I}}_{m+1,p}$ have dimensions $(m+1) \times p$, where $p < (m+1)$, and $p < n$. The truncation size $p$ depends on the information content in the measurement, with typically $2 \leq p \leq 4$ for GHG measurements described here. The choice of $p$ will be discussed in more detail in section 5.

We note that the truncation of $\mathbf{V}$ leads to the matrix multiplication of $\tilde{\mathbf{V}}$ and $\tilde{\mathbf{V}}^T$ being non-commutative for the general case:

$$
\tilde{\mathbf{V}}^T \tilde{\mathbf{V}} = \mathbf{I}_p,
\tag{10}
$$

$$
\tilde{\mathbf{V}} \tilde{\mathbf{V}}^T \neq \mathbf{I}_{m+1}.
\tag{11}
$$

The subscript to $\mathbf{I}$ denotes the dimensions of the identity matrix. This non-commutative behavior has implications on the types of biases resulting from the SVD truncation as we will later see in section 3.5.

Finally, for completeness, we will look at transformations between the $\mathbf{x}$ and $\mathbf{z}$ bases. Given a vector $\mathbf{z}$, one can project it back on to the $\mathbf{x}$-basis using

$$
\mathbf{x} = \mathbf{V} \tilde{\mathbf{I}}_{m+1,p} \mathbf{z}.
\tag{12}
$$

However, conversion using Eq. (12) only projects on to a subspace of $\mathbf{x}$. Mathematically, in making a transformation from $\mathbf{x}$ to $\mathbf{z}$ using Eq. (9) and back to $\mathbf{x}$ using Eq. (12), any information corresponding to the $m+1-p$ dimensions not present in the $\mathbf{z}$ basis is lost. But, starting with the reduced basis space $\mathbf{z}$, one can transform to the $\mathbf{x}$-basis and back with no loss of information.





### 3.2 Retrieval equations in the z basis

By substituting Eq. (12) into Eq. (7), effectively projecting the retrieval onto the subspace of $\mathbf{x}$ spanned by $\mathbf{z}$, one can solve the retrieval equation:

$$\mathbf{K}^T\mathbf{S}_\epsilon^{-1}\mathbf{y} = \mathbf{K}^T\mathbf{S}_\epsilon^{-1}\mathbf{K}\tilde{\mathbf{V}}\hat{\mathbf{z}}$$
$$\tilde{\mathbf{V}}^T\mathbf{K}^T\mathbf{S}_\epsilon^{-1}\mathbf{y} = \tilde{\mathbf{V}}^T\mathbf{K}^T\mathbf{S}_\epsilon^{-1}\mathbf{K}\tilde{\mathbf{V}}\hat{\mathbf{z}}.$$

In the second equation line above, we have multiplied both sides by $\tilde{\mathbf{V}}^T$ to also reduce the column space of the equation to the $\mathbf{z}$ basis. This yields an estimate $\hat{\mathbf{z}}$ in the $\mathbf{z}$ basis,

$$\hat{\mathbf{z}} = \mathbf{G}_{\text{SVD}}\mathbf{y}, \quad \text{where} \tag{13}$$
$$\mathbf{G}_{\text{SVD}} = \left[\tilde{\mathbf{V}}^T\mathbf{K}^T\mathbf{S}_\epsilon^{-1}\mathbf{K}\tilde{\mathbf{V}}\right]^{-1}\tilde{\mathbf{V}}^T\mathbf{K}^T\mathbf{S}_\epsilon^{-1}. \tag{14}$$

$\mathbf{G}_{\text{SVD}}$ is a $p \times n$ matrix analogous to the $\mathbf{G}$ or "Gain" matrix used in (Rodgers, 2000). In determining $\mathbf{G}_{\text{SVD}}$, one needs to ensure sufficient truncation in $\tilde{\mathbf{V}}$ to ensure that the $p \times p$ matrix $[\tilde{\mathbf{V}}^T\mathbf{K}^T\mathbf{S}_\epsilon^{-1}\mathbf{K}\tilde{\mathbf{V}}]$ is invertible. Since $\tilde{\mathbf{V}}$ consists of the eigenvectors of $\mathbf{K}^T\mathbf{S}_\epsilon^{-1}\mathbf{K}$, truncation can easily be done by selecting only eigenvectors with positive eigenvalues.

Equation (13), by selecting just the principal components, offers a way of regularizing and solving Eq. (7) without relying on the assumption of a prior distribution in $\mathbf{x}$. This allows an alternative retrieval method to the commonly used Bayesian optimal estimation method.

### 3.3 Relationship between SVD and OE retrieval

In this section we will explicitly describe the SVD retrieval as an OE retrieval with a particular uninformative prior and with the replacing of the inverse with the pseudo-inverse in computing the gain matrix $\mathbf{G}_{\text{OE}}$. Although the algebra here has been shown previously (Rodgers, 2000; Butz et al., 2010), we find it useful to think of the SVD method as simply implementing a pseudo-inverse in lieu of an inverse to solve the underdetermined retrieval equations.

We start with the analogous traditional OE version of (13) as described in Rodgers (2000):

$$\mathbf{x}_{\text{OE}} = \mathbf{x}_a + \mathbf{G}_{\text{OE}}(\mathbf{y} - \mathbf{F}(\mathbf{x}_a)), \tag{15}$$

where

$$\mathbf{G}_{\text{OE}} = (\mathbf{S}_a^{-1} + \mathbf{K}^T\mathbf{S}_\epsilon^{-1}\mathbf{K})^{-1}\mathbf{K}^T\mathbf{S}_\epsilon^{-1}$$

where $\mathbf{x}_a$ and $\mathbf{S}_a$ are the *a priori* mean and covariance matrix of the state vector $\mathbf{x}$, respectively. With no loss in generality, we set $\mathbf{x}_a = 0$. We then use an uninformative prior where $\mathbf{S}_a$ is infinitely large such that $\mathbf{S}_a^{-1} = 0$. Note that this prior contains no information on the distribution of the state $\mathbf{x}$, hence the name 'uninformative' prior. The above equations then reduce to the




following:

$$\mathbf{x}_{\mathrm{OE}} = \mathbf{G}_{\mathrm{OE}}\mathbf{y} \tag{16}$$

$$\mathbf{G}_{\mathrm{OE}} = (\mathbf{K}^T\mathbf{S}_\epsilon^{-1}\mathbf{K})^{-1}\mathbf{K}^T\mathbf{S}_\epsilon^{-1}. \tag{17}$$

Without the term $\mathbf{S}_a^{-1}$ in (17), $\mathbf{K}^T\mathbf{S}_\epsilon^{-1}\mathbf{K}$ might not be full-rank and hence non-invertible. We will replace its inverse with the pseudo-inverse, which is well defined even for singular matrices. The gain matrix is now

$$\mathbf{G}_{\mathrm{OE}} = (\mathbf{K}^T\mathbf{S}_\epsilon^{-1}\mathbf{K})^{+}\mathbf{K}^T\mathbf{S}_\epsilon^{-1}, \tag{18}$$

where the superscript $^+$ above a matrix denotes its pseudo-inverse. Since $\mathbf{S}_\epsilon$ is positive definite, $\mathbf{K}^T\mathbf{S}_\epsilon^{-1}\mathbf{K}$ is positive-semidefinite and its singular value decomposition is identical to its eigenvalue decomposition. Therefore, we can express the singular value decomposition of $\mathbf{K}^T\mathbf{S}_\epsilon^{-1}\mathbf{K}$ as follows:

$$\mathbf{K}^T\mathbf{S}_\epsilon^{-1}\mathbf{K} = \mathbf{V}\mathbf{D}\mathbf{V}^T \tag{19}$$

where $\mathbf{\Gamma}^T\mathbf{\Gamma} = \mathbf{D}$, a $(m+1)\times(m+1)$ diagonal matrix. $\mathbf{\Gamma}$ here is the same as defined in Eq. (8). We can truncate the right hand side of the equation to remove degenerate rows in $\mathbf{V}$ and degenerate rows and columns in $\mathbf{D}$, without affecting the equality. However, we choose to truncate further to rank $p$ to get numerical stability:

$$\mathbf{K}^T\mathbf{S}_\epsilon^{-1}\mathbf{K} \approx \tilde{\mathbf{V}}\tilde{\mathbf{D}}\tilde{\mathbf{V}}^T, \tag{20}$$

where $\tilde{\mathbf{V}}$ and $\tilde{\mathbf{D}}$ are truncated versions, with $\tilde{\mathbf{V}}$ being identical to that used in Eq. (9). If the truncation is applied only to the degenerate rows, the approximation in Eq. (20) can be replaced by equality and the pseudo-inverse can be constructed from the singular value decomposition per Petersen and Pedersen (2012). The key results still hold even when we choose to more aggressively truncate $\mathbf{D}$ to rank $p$ in our SVD method. This is equivalent to replacing the term $\mathbf{K}^T\mathbf{S}_\epsilon^{-1}\mathbf{K}$ in (17) with the closest rank-$p$ matrix approximation under the Frobenious norm and then computing its pseudo-inverse (Eckart and Young, 1936):

$$(\mathbf{K}^T\mathbf{S}_\epsilon^{-1}\mathbf{K})^{+} = \tilde{\mathbf{V}}\tilde{\mathbf{D}}^{-1}\tilde{\mathbf{V}}^T. \tag{21}$$

Substituting $\tilde{\mathbf{D}}$ from (20) into (21), we see that

$$\begin{aligned}\mathbf{G}_{\mathrm{OE}} &= (\mathbf{K}^T\mathbf{S}_\epsilon^{-1}\mathbf{K})^{+}\mathbf{K}^T\mathbf{S}_\epsilon^{-1} \\ &= \tilde{\mathbf{V}}(\tilde{\mathbf{V}}^T\mathbf{K}^T\mathbf{S}_\epsilon^{-1}\mathbf{K}\tilde{\mathbf{V}})^{-1}\tilde{\mathbf{V}}^T\mathbf{K}^T\mathbf{S}_\epsilon^{-1} \\ &= \tilde{\mathbf{V}}\mathbf{G}_{\mathrm{SVD}}.\end{aligned} \tag{22}$$

The result in Eq. (22) indicates that the OE retrieval with $\mathbf{x}_a = 0$ and $\mathbf{S}_a^{-1} = 0$ is, up to a linear transformation, identical to an SVD retrieval where we truncate the basis vectors $\mathbf{V}$. In other words, the SVD retrieval may be viewed as a special case of the OE retrieval that uses an uninformative prior for the state $\mathbf{x}$. This has also been found by Cressie et al. (2017) in their analysis of the AIRS retrieval algorithm.





### 3.4 SVD Retrieval Error Covariance Matrix and Averaging Kernels

One of the strengths of the OE method is the ability to propagate errors from the inputs to the final estimate of the state vector $\mathbf{x}$. Given the prior covariance matrix $\mathbf{S}_a$ and measurement-error covariance matrix $\mathbf{S}_\epsilon$, Rodgers (2000) demonstrated that the posterior covariance matrix for the OE estimate in Eq. (16) is,

$$\mathbf{S_{x,oe}} = \left(\mathbf{S}_a^{-1} + \mathbf{K}^T\mathbf{S}_\epsilon^{-1}\mathbf{K}\right)^{-1}. \tag{23}$$

Since we have demonstrated in Section 3.3 that the SVD approach is equivalent to optimal estimation with $\mathbf{x}_a = 0$ and $\mathbf{S}_a^{-1} = 0$, we can apply those values into (23) to obtain the SVD posterior covariance matrix $\mathbf{S_{x,svd}}$ as follows,

$$\mathbf{S_{x,svd}} = \left(\mathbf{K}^T\mathbf{S}_\epsilon^{-1}\mathbf{K}\right)^{-1}. \tag{24}$$

In some application, $\left(\mathbf{K}^T\mathbf{S}_\epsilon^{-1}\mathbf{K}\right)$ might not be full rank, and thus the expression in (24) may be approximated using a pseudo-inverse:

$$\mathbf{S_{x,svd}} = \left(\mathbf{K}^T\mathbf{S}_\epsilon^{-1}\mathbf{K}\right)^{+}. \tag{25}$$

Note that the SVD posterior matrix in Eq. (24) is in the $\mathbf{x}$-basis. It is straightforward to transform it to the $\mathbf{z}$-basis using the linear transformation in Eq. (9):

$$
\begin{aligned}
\mathbf{S_z} &= \tilde{\mathbf{V}}^T\mathbf{S_{x,svd}}\tilde{\mathbf{V}} \\
&= \tilde{\mathbf{V}}^T\left(\mathbf{K}^T\mathbf{S}_\epsilon^{-1}\mathbf{K}\right)^{+}\tilde{\mathbf{V}} \\
&= \tilde{\mathbf{V}}^T\tilde{\mathbf{V}}\left(\tilde{\mathbf{V}}^T\mathbf{K}^T\mathbf{S}_\epsilon^{-1}\mathbf{K}\tilde{\mathbf{V}}\right)^{-1}\tilde{\mathbf{V}}^T\tilde{\mathbf{V}} \\
\mathbf{S_z} &= \left(\tilde{\mathbf{V}}^T\mathbf{K}^T\mathbf{S}_\epsilon^{-1}\mathbf{K}\tilde{\mathbf{V}}\right)^{-1} \\
&= \left(\tilde{\mathbf{I}}_{m+1,p}^T\,\mathbf{\Gamma}^T\mathbf{\Gamma}\,\tilde{\mathbf{I}}_{m+1,p}\right)^{-1}.
\end{aligned} \tag{26}
$$

Since $\mathbf{\Gamma}$ and $\tilde{\mathbf{I}}_{m+1,p}$ have non-zero elements only on the main diagonal, the retrieval error covariance matrix $\mathbf{S_z}$ has no off-diagonal terms implying that errors in the retrieved parameters are uncorrelated.

The averaging kernels of the $\mathbf{z}$ retrieval elements can be calculated using (Eskes and Boersma, 2003)

$$
\begin{aligned}
\mathbf{A}_{\text{SVD}} &= \frac{\partial\hat{\mathbf{z}}}{\partial\mathbf{x}} = \mathbf{G}_{\text{SVD}}\mathbf{K} \\
&= \left[\tilde{\mathbf{V}}^T\mathbf{K}^T\mathbf{S}_\epsilon^{-1}\mathbf{K}\tilde{\mathbf{V}}\right]^{-1}\tilde{\mathbf{V}}^T\mathbf{K}^T\mathbf{S}_\epsilon^{-1}\mathbf{K}, \\
\mathbf{A}_{\text{SVD}}\tilde{\mathbf{V}} &= \left[\tilde{\mathbf{V}}^T\mathbf{K}^T\mathbf{S}_\epsilon^{-1}\mathbf{K}\tilde{\mathbf{V}}\right]^{-1}\tilde{\mathbf{V}}^T\mathbf{K}^T\mathbf{S}_\epsilon^{-1}\mathbf{K}\tilde{\mathbf{V}} \\
&= \mathbf{I}_p \quad p \times p \text{ identity matrix} \\
\mathbf{A}_{\text{SVD}} &= \tilde{\mathbf{V}}^T.
\end{aligned} \tag{27}
$$

$$\mathbf{A}_{\text{SVD}} = \tilde{\mathbf{V}}^T. \tag{28}$$

Thus, in calculating the averaging kernel, one obtains the simplified Eq. (28), where $\tilde{\mathbf{V}}^T$ can be directly obtained from the singular value decomposition step with the appropriate truncation. The reader should note that the degree of truncation only



affects the number of components (rows) in the averaging kernel $\mathbf{A}_{\text{SVD}}$, but not the information content (number of columns) or shape of the individual components themselves.

### 3.5 Bias-free nature of SVD principal components

In practice, biases in GHG measurements occur due to several reasons, many of which are out of the scope of this paper. To limit the discussion to biases arising from the solving of an underdetermined problem using some form of regularization (retrieval error, which is universal to all GHG measurements) we make two further assumptions:

1. Negligible error from imperfect knowledge of the atmospheric pressure, temperature and water vapor profile (**b**). These errors have been found to be small in practice (Abshire et al., 2017) and can be further reduced with auxiliary measurements.

2. Negligible errors in radiative transfer equations (forward model), instrument calibration or other similar systematic effects.

Retrieval errors in the traditional OE method arise from incorrect assumptions about the true greenhouse gas profile distribution. For the SVD method, we see a potential bias in retrievals in the original $\mathbf{x}$ basis, but not in the principal component $\mathbf{z}$ basis.

We will first derive the expected bias for the OE method where the input prior mean and covariance matrix are incorrect. We assume that true state process $\mathbf{x}$ has the true mean $\mathbf{x}_t$ and the true covariance matrix $\mathbf{S}_t$. However, we assume that in practice the OE algorithm is using the prior mean $\mathbf{x}_a$ and covariance matrix $\mathbf{S}_a$. Note that true prior distribution, $\{\mathbf{x}_t, \mathbf{S}_t\}$, and the one used in the computations, $\{\mathbf{x}_a, \mathbf{S}_a\}$, are not necessarily the same. When they do differ, there is an expected bias, which we will show below.

The expected bias is defined as

$$\text{Bias} = E(\hat{\mathbf{x}} - \mathbf{x}_t), \tag{29}$$

where $E()$ denotes the expectation value averaged over several measurements, such that random noise, $\epsilon$ averages out to zero. We now substitute the OE retrieval equation (Eq. 15:

$$\begin{aligned} \text{Bias}_{\text{OE}} &= E(\mathbf{x}_a + \mathbf{G}_{\text{OE}}\mathbf{y} - \mathbf{x}_t), \\ &= E(\mathbf{x}_a + \mathbf{G}_{\text{OE}}(\mathbf{K}(\mathbf{x}_t - \mathbf{x}_a) + \epsilon) - \mathbf{x}_t), \end{aligned}$$

where we have applied the forward model equation for a true state $\mathbf{x}_t$ with noise $\epsilon$. Simplifying the equation and substituting for $\mathbf{G}_{\text{OE}}$, we get

$$\begin{aligned} \text{Bias}_{\text{OE}} &= (\mathbf{I} - \mathbf{G}_{\text{OE}}\mathbf{K})(\mathbf{x}_a - \mathbf{x}_t) \\ &= (\mathbf{S}_a^{-1} + \mathbf{K}\mathbf{S}_\epsilon^{-1}\mathbf{K})^{-1}\mathbf{S}_a^{-1}(\mathbf{x}_a - \mathbf{x}_t) \end{aligned} \tag{30}$$

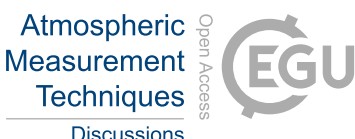

Looking at Eq. (30), we see that the expected bias in OE retrievals is proportional to $(\mathbf{x}_a - \mathbf{x}_t)$, or the difference between true prior mean and one used in practice. Eq. (30) also shows that when the constraint on the Bayesian OE is set too high ($\mathbf{S}_a$ is small and thus $\mathbf{S}_a^{-1}$ is large), there is a significant bias in the retrieval from the mismatch between the true mean and the prior mean assumed in the retrieval.

We now derive the bias when using the SVD method. We assume a true state $\mathbf{x}_t$ and an uninformative prior $\mathbf{x}_u$. We again start with the bias equation

$$\text{Bias} = E(\hat{\mathbf{x}} - \mathbf{x}_t),$$

where $\hat{\mathbf{x}}$ denotes the SVD retrieved result $\hat{\mathbf{z}}$ (Eq. 13) transformed to the $\mathbf{x}$ basis using Eq. 12. This can be expanded to give

$$\text{Bias}_{\text{SVD}} = E(\mathbf{x}_u + \tilde{\mathbf{V}}\mathbf{G}_{\text{SVD}}\mathbf{y} - \mathbf{x}_t). \tag{31}$$

We have deliberately left in the uninformative prior $\mathbf{x}_u$ for better illustration of the bias. We now include the linearized forward model (Eq. 4), and the noise (Eq. 2):

$$
\begin{aligned}
\text{Bias}_{\text{SVD}} &= E(\mathbf{x}_u + \tilde{\mathbf{V}}\mathbf{G}_{\text{SVD}}(\mathbf{K}(\mathbf{x}_t - \mathbf{x}_u) + \epsilon) - \mathbf{x}_t). && (32) \\
&= (\mathbf{I} - \tilde{\mathbf{V}}\mathbf{G}_{\text{SVD}}\mathbf{K})(\mathbf{x}_u - \mathbf{x}_t) \\
&= (\mathbf{I} - (\mathbf{K}^T\mathbf{S}_\epsilon^{-1}\mathbf{K})^+\mathbf{K}^T\mathbf{S}_\epsilon^{-1}\mathbf{K})(\mathbf{x}_u - \mathbf{x}_t), && (33)
\end{aligned}
$$

where we have applied the pseudo-inverse derivation (Eqs. 18 and 22) of the SVD retrieval.

As we can see in Eq. (33), when the term $\mathbf{K}^T\mathbf{S}_\epsilon^{-1}\mathbf{K}$ is singular, then the product of it against its pseudoinverse (i.e., $(\mathbf{K}^T\mathbf{S}_\epsilon^{-1}\mathbf{K})^+\mathbf{K}^T\mathbf{S}_\epsilon^{-1}\mathbf{K}$) is not equal to the identity matrix, and hence the bias will generally be non-zero. This can be further illustrated by applying Eqs. (19) and (21),

$$\text{Bias}_{\text{SVD}} = (\mathbf{I} - \tilde{\mathbf{V}}\tilde{\mathbf{V}}^T)(\mathbf{x}_u - \mathbf{x}_t), \tag{34}$$

where $\tilde{\mathbf{V}}\tilde{\mathbf{V}}^T$ is of rank $p$ and not equal to the $(m{+}1)$ rank identity matrix (see Eq. 11) .

Fortunately, when we look at the retrievals on the $\mathbf{z}$ space, the retrievals are unbiased. Given that there is no loss of information in projecting the retrieval results from the $\mathbf{z}$ basis to the $\mathbf{x}$ basis as was done above, we can simply project Eq. 34 back to the $\mathbf{z}$ basis using Eq. 9:

$$
\begin{aligned}
\text{Bias}_{\text{SVD},\mathbf{z}} &= \tilde{\mathbf{V}}^T(\mathbf{I} - \tilde{\mathbf{V}}\tilde{\mathbf{V}}^T)(\mathbf{x}_u - \mathbf{x}_t), \\
&= (\tilde{\mathbf{V}}^T - \tilde{\mathbf{V}}^T\tilde{\mathbf{V}}\tilde{\mathbf{V}}^T)(\mathbf{x}_u - \mathbf{x}_t), \\
&= 0, && (35)
\end{aligned}
$$

where we have used $\tilde{\mathbf{V}}^T\tilde{\mathbf{V}} = \mathbf{I}_p$ from Eq. 10 . It should be noted that the bias free result holds *regardless* of the degree of truncation or choice of the uninformative prior. The bias free result will be illustrated via numerical simulations in section 5.





### 3.6 SVD Retrieval Validation

SVD retrievals can be validated directly in the retrieval $\mathbf{z}$-basis by transforming the validation data in the parameter space $\mathbf{x}$-basis using (9). Since the retrieval error covariance matrix $\mathbf{S_z}$ is defined in this basis, the expected scatter based on the assumed noise distribution, calculated using (26) can be compared against the actual scatter based on a large number of measurements.

### 4 SVD approach applied to IPDA lidar $CO_2$ measurements

We choose the NASA Goddard $CO_2$ Sounder instrument concept (Abshire et al., 2014) as an example to describe the SVD technique. The $CO_2$ Sounder is a lidar instrument that probes the 1572.335 nm $CO_2$ absorption line with multiple (between 15 and 30) wavelength samples (Abshire et al., 2017). To best illustrate the matrix algebra, we choose the 15-wavelength sampling scheme from a recent field campaign ($n = 15$). The column absorption lineshape and wavelength sampling are shown in Figure 1. For $CO_2$, the atmosphere can be modeled using 100 layers ($m = 100$), where the layers are spaced almost evenly in pressure to have equal weight.

### 4.1 Forward model

The forward model can be linearized to produce the Kernel matrix $\mathbf{K}$ as shown in (5). In Figure 2, we illustrate $\mathbf{K}$ by plotting two columns of the $\mathbf{K}$ kernel matrix and two rows. The heterogeneity of the $\mathbf{K}$ matrix in row and column space is key for the SVD technique to be able to extract principal components.

### 4.2 Measurement Noise

For IPDA lidar instruments, one primary limitation is photon shot noise, which is a fundamental quantum noise with variance equal to the number of photons detected. Photon shot noise is the key limiting factor of measurement precision when lidar instruments are laser power limited, which is often the case. Although other forms of noise such as detector dark current noise, laser speckle noise and solar background noise also play a role, their effect on the principal components is limited. For this reason and for simplicity, we will assume a photon shot noise limited lidar instrument.

For this example, we will assume an integrated photon count of $s_0$ for a wavelength sample with no $CO_2$ absorption. This would give an optical signal level of

$$s(\lambda_j) = s_o T(\lambda_j, \mathbf{x}) \tag{36}$$

for each of the wavelength samples $\lambda_j$. From the definition of the forward model in (3), we can set

$$
\begin{aligned}
y_j &= -\log\left(\frac{s(\lambda_j)}{T(\lambda_j, \mathbf{x_0})}\right) \\
\langle (y_j - \langle y_j \rangle)^2 \rangle &= \left(\frac{1}{s(\lambda_j)} \times \sqrt{s(\lambda_j)}\right)^2 \\
&= \frac{1}{s_0 T(\lambda_j, \mathbf{x})}
\end{aligned}
\tag{37}
$$





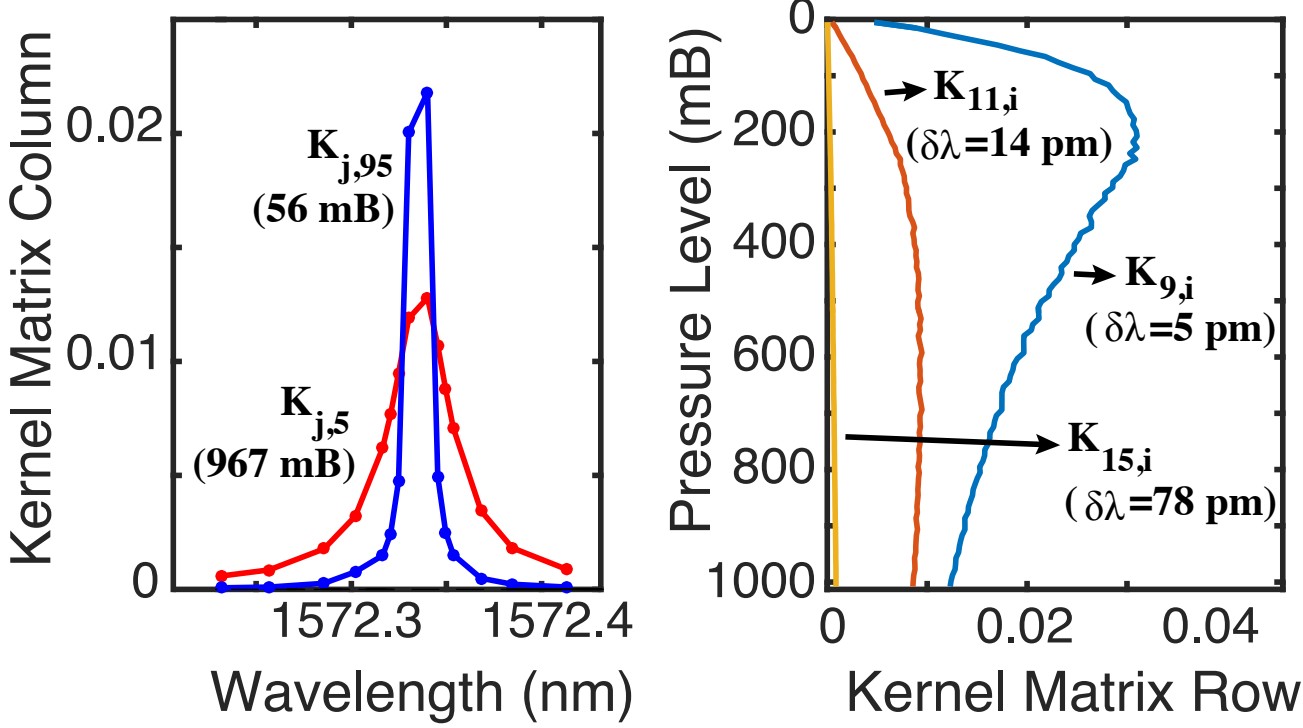

**Figure 2.** (Color online) Forward model $\mathbf{K}$ matrix : (left) We illustrate the $\mathbf{K}$ matrix by plotting two columns, each corresponding to the absorption due to a certain slice of the atmosphere. With increasing atmospheric pressure, the absorption lineshape is pressure-broadened. (right) We plot three rows of $\mathbf{K}$, each showing the dependence of the absorption to different parts of the atmosphere for a given measurement wavelength sample. The sample wavelength corresponding to each row has been expressed as a deviation from the absorption line center 1572.335122 nm. The absorption is lower the further one deviates from the absorption line center.

Using (37), we can now define the measurement error covariance matrix as

$$\mathbf{S}_\epsilon = \frac{1}{s_0} \begin{bmatrix} T(\lambda_1,\mathbf{x})^{-1} & 0 & \dots & 0 \\ 0 & T(\lambda_2,\mathbf{x})^{-1} & \dots & 0 \\ \vdots & \vdots & \ddots & \vdots \\ 0 & 0 & \dots & T(\lambda_n,\mathbf{x})^{-1} \end{bmatrix} \tag{38}$$

### 4.3 SVD Averaging Kernels and Error Covariance

The $CO_2$ column dependence of the averaging kernels are plotted in Figure 3. The first term of the singular value decomposition
5  is mostly derived from $x_0$ ($\tilde{\mathbf{V}}_{1,1} > 0.999$), and thus a measure of the mean signal amplitude (see Figure 3, left). It is more sensitive to wavelength samples on the wings of the absorption line since those do not see much of $CO_2$ absorption. The





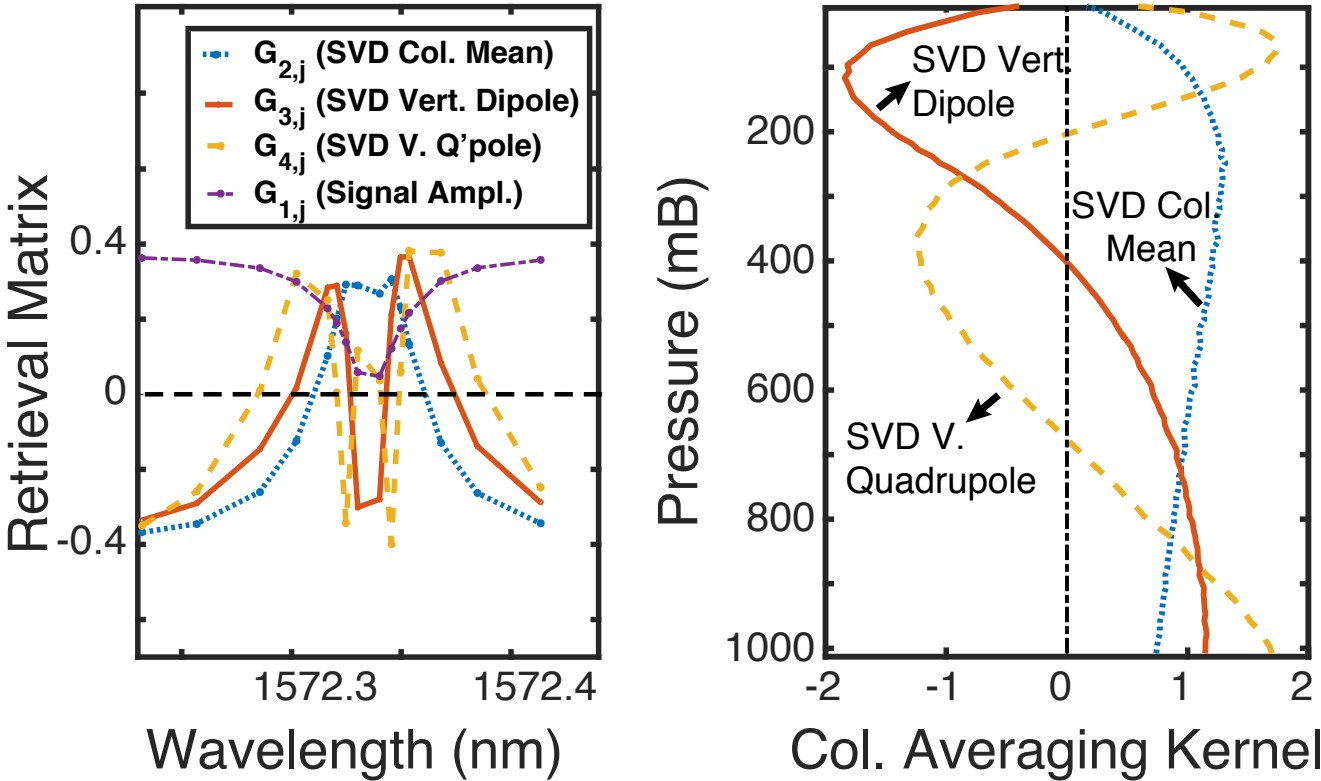

**Figure 3.** (Color online) SVD retrieval basis: (left) The rows of the **G** matrix are plotted as a function of the wavelength of the measurement samples. (right) The averaging kernels of the first three $CO_2$ principal component terms are plotted. Each subsequent term has an additional zero-crossing in the averaging kernel.

second term of the SVD is the first $CO_2$ principal component (PC) and behaves like a column-averaged $CO_2$ mixing ratio or $X_{CO2}$ with units of ppm.

The third term from the SVD or second $CO_2$ PC behaves analogous to a dipole moment (for instance, the electric dipole moment in physics) and can be assigned dipole moment units of ppm $B^2$. Analogous to the electric dipole moment, which has

5 units of electric charge $\times$ distance, this dipole moment has units of ppm B ("charge") $\times$ B ("distance") = ppm $B^2$.

The vertical dipole PC carries information about the vertical distribution of $CO_2$ and will be examined in detail in the next section. Typical values are between -0.5 to 0.5 ppm $B^2$, with more extreme values going up to $\pm 1.5$ ppm $B^2$. One requires a precision of about 0.1 ppm $B^2$ in retrieving the $CO_2$ vertical dipole moment in order to provide some useful vertical information about the $CO_2$ distribution. The fourth term or third $CO_2$ PC is the quadrupole moment of the column $CO_2$ mixing ratio profile.

10 The averaging kernels for the first three $CO_2$ principal components are plotted in Figure 3 (right).

As seen earlier in (26), the SVD approach ensures that the random error in the retrieved quantities are uncorrelated. The variance of the retrieved quantities increases with principal component order, with the vertical dipole moment being about 4.5





times less precise (standard deviation) than the column mean and the vertical quadrupole moment being a further 7 times less precise.

### 4.4 Effect of spectral resolution on SVD retrievals

Higher order $CO_2$ PCs rely on the differential pressure broadening in the $CO_2$ absorption lineshape along the atmospheric
column to provide information about the vertical distribution of $CO_2$. For this reason, unlike the column $X_{CO2}$, they are expected to be sensitive to the instrument spectral resolution.

For passive spectrometers that work by resolving sunlight passing through the atmosphere, the instrument spectral resolution and sampling resolution are directly related, and often close to one another. In contrast, lidar instruments probe the atmosphere with essentially monochromatic light, *i.e.* the laser spectral width is much narrower than gas absorption linewidth, and have
spectral resolutions orders of magnitude smaller than the sampling resolution. It is important for the reader to note that high quality measurements for the purposes of obtaining vertical information require high spectral resolution but not necessarily high sampling resolution. As we shall see in this section, for a given sampling, the measurement precision depends strongly on the instrument spectral resolution.

We calculate the expected random noise in retrieving the first two PCs for a range of instrument spectral resolutions (see
Figure 4). The precision of the column $X_{CO2}$ showed little change with poorer spectral resolution as one would expect. In contrast, the precision of the $CO_2$ vertical dipole moment very quickly degrades with instrument line broadening. The result has been calculated for the specific case of a wavelength sampling scheme sampling a single absorption line, *i.e.* the $CO_2$ Sounder lidar sampling scheme. Nevertheless, the results still give some indication on the importance of spectral resolution.

The ability of satellite-based passive spectrometers to resolve the $CO_2$ vertical structure is expected to be significantly
hampered by their poorer spectral resolutions compared to instruments like TCCON or the $CO_2$ Sounder. Although in theory, random error can be overcome by longer integration times or having more wavelength samples, in practice, as has been the experience of satellite GHG instruments as of the time of writing, systematic effects ultimately limit the accuracy of the measurement. Thus, spectral resolution is crucial in trying to resolve information on the vertical GHG distribution.

### 5   Numerical Simulations comparing Singular Value Decomposition with Bayesian Optimal Estimation

In this section, we will look at the retrieval performance of the SVD and traditional OE (refers to OE retrievals with finite $\mathbf{S}_a$) methods for simulated data. After describing the methodology used for comparisons, we will highlight the pitfalls of using too strong or too weak a constraint with the traditional OE method, and how the SVD method provides useful information in the principal component basis independent of the degree of constraint. Then, we will show a case where the SVD approach successfully extracts vertical $CO_2$ information from the absorption measurement.





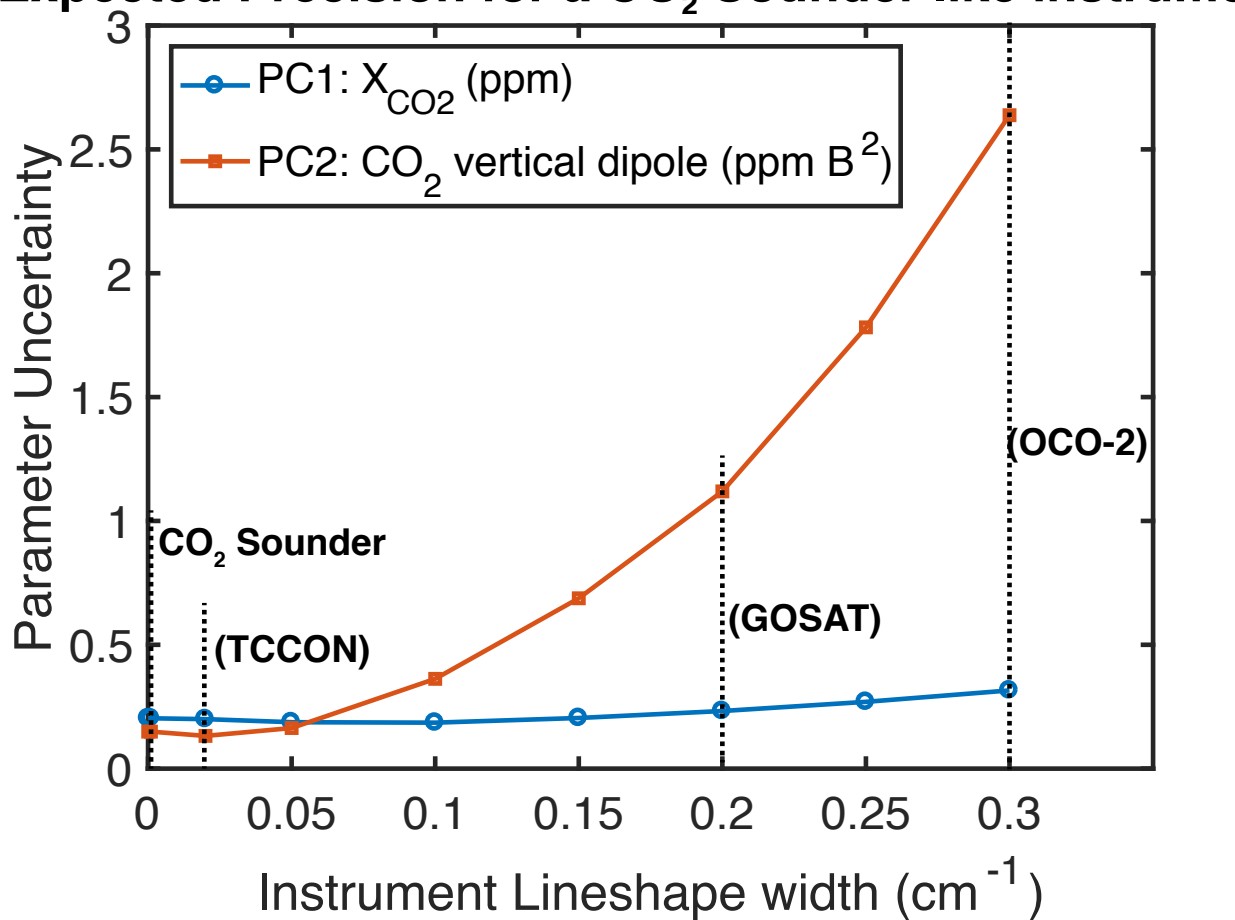

**Figure 4.** (Color online) Retrieval precision versus instrument spectral linewidth for the first two $CO_2$ principal components (PCs) : While the column $X_{CO2}$ is largely unaffected by the spectral resolution, the precision of the $CO_2$ vertical dipole moment degrades strongly with poorer resolution. We assume a $CO_2$ instrument model, but with some instrument line broadening. The x-axis denotes the full-width at half maximum of the triangular instrument line shape used to broaden the $CO_2$ absorption. We assume photon shot noise with a SNR of 1000 for points with no $CO_2$ absorption. The spectral resolutions of TCCON Wunch et al. (2011), GOSAT Kuze et al. (2009), OCO-2 Connor et al. (2008) and $CO_2$ Sounder instruments are indicated, though the calculations done in this work apply only to the $CO_2$ Sounder instrument.



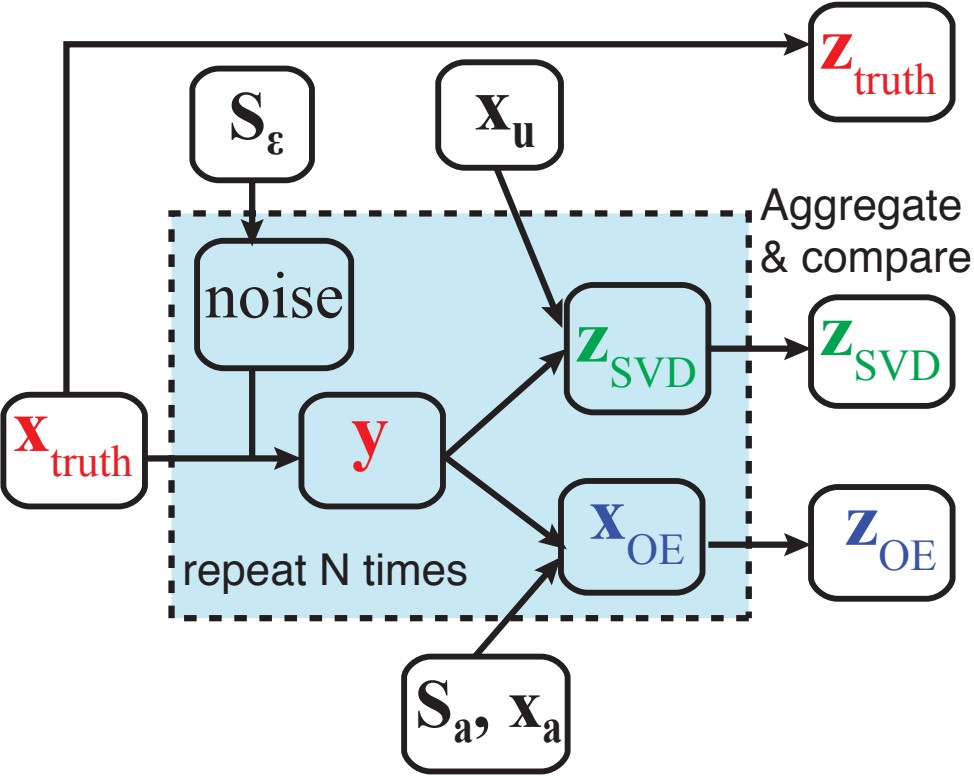

**Figure 5.** (Color online) Comparing SVD and OE retrievals : Using simulated data $\mathbf{y}$ generated from a $CO_2$ profile $\mathbf{x}_{\text{truth}}$, we perform SVD and traditional OE retrievals and compare their results averaged over an N=1000 ensemble projected onto the $\mathbf{z}$-basis. Specifically, we look at the variance and the bias compared to $\mathbf{z}_{\text{truth}}$, which is projected from $\mathbf{x}_{\text{truth}}$. We specify an uninformative prior $\mathbf{x}_u$ for the SVD results and a Bayesian prior mean $\mathbf{x}_a$ and prior covariance matrix $\mathbf{S}_a$.

## 5.1 Methodology

For the simulations, we use a $CO_2$ Sounder instrument model with 30 wavelength samples ($n = 30$) to better illustrate the shape of the residuals. The measurement is made over a vertical air column from the surface to the top-of-atmosphere. For the full model basis (x-basis), we divide the atmosphere into 100 equal levels ($m = 100$), each spanning a 10 mB pressure interval. We make comparisons for three different cases, which will be described in the following subsections.

For each case, we define a "true" $CO_2$ profile, $\mathbf{x}_{\text{truth}}$ and compute the total absorption lineshape. We then compute the signal at the sample wavelengths (using (36)) and add photon shot noise as per (37) to the create a "measurement" $\mathbf{y}$ (see Figure 5. For all simulations, we set $s_0 = 10^6$, which implies SNR=1000 for points with no $CO_2$ absorption. The measurement error covariance matrix is computed using (38). We then perform retrievals with the traditional OE (using (15)) and SVD (using



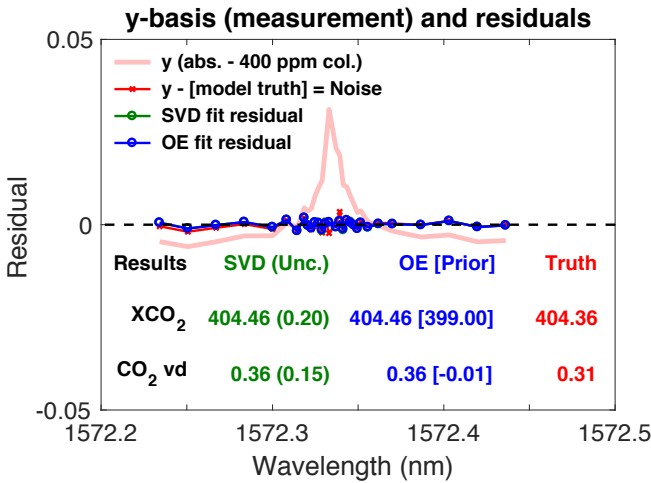
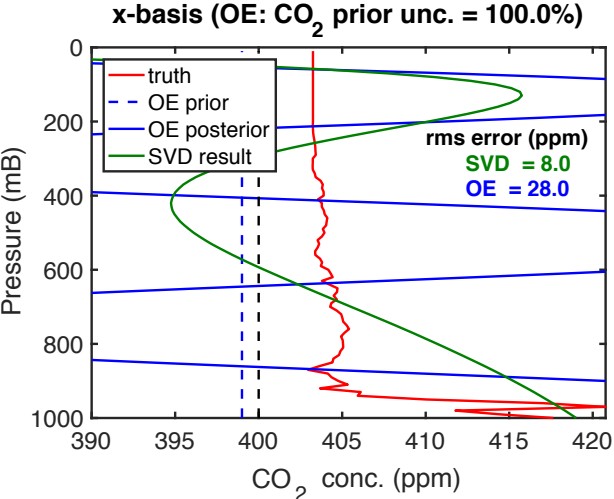

**Figure 6.** (Color online) Sample retrieval for a single simulated measurement (noise instance) under weak constraint (4 principal components for SVD, 100% prior uncertainty for each $CO_2$ level for OE) - (left) The SVD and traditional OE approaches successfully minimize the fit residual to match that of the noise, thus demonstrating convergence. Results projected to the **z**-basis show reasonable performance of the $X_{CO2}$ column mean (first principal component), but poor performances for higher order terms (not shown), indicating overfitting to the noise. (right) Results projected to the **x**-basis show highly oscillatory and divergent profiles due to the instability in overfitting. Thus, traditional OE results in the full model **x**-basis are not useful, and need a projection onto the **z**-basis or other transformation. Note - This has been shown for illustrative purposes. A proper evaluation of the methods requires an ensemble average of such simulations (see Fig. 8 left and center)

(13)) approaches in their respective bases. By doing this for an ensemble of "measurements", using the same $\mathbf{x}_{\text{truth}}$ and $s_0$ but a different instance of noise each time, we get a set of results that can be characterized by a mean and standard deviation.

Since the SVD principal components are unbiased, we make quantitative comparisons between the two techniques in that **z**-basis. While the idea of using the **z**-basis might seem new, in practice, column-averaged measurements are typically used for

5  flux estimations. Thus, results in the **z**-basis for the OE method do have wider implications. We also project the SVD results back on to the **x**-basis using (12) to get an intuitive sense of how the SVD and OE approaches work. This last projection is essentially a reduced-rank pseudo-inverse calculation (see section 3.3).

For the SVD approach, we set the uninformative prior $\mathbf{x}_u$ to be a uniform 400 ppm $CO_2$ profile. For the OE approach, the Bayesian prior mean is kept simple and chosen on a case-by-case basis. For the prior covariance, we assume a 200 mB $1/e^2$

10  vertical correlation distance in the $CO_2$ concentration in the atmosphere.

## 5.2 Constraining the retrieval for regularization

GHG retrievals require some sort of constraint to regularize the retrieval problem (see section 2.1). The level of constraint of an OE retrieval can be expressed as the relative strength of the weighting on the prior value, which is inversely proportional to





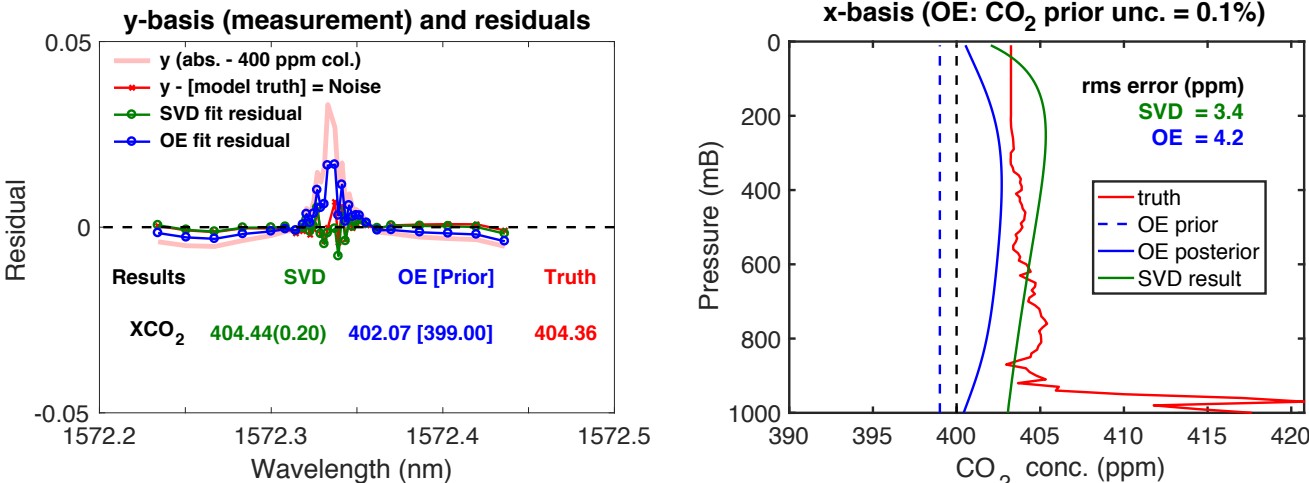

**Figure 7.** (Color online) Sample retrieval for a single simulated measurement under strong constraint (1 principal component for SVD, 0.1% prior uncertainty for each $CO_2$ level for OE) - (left) Both the SVD and traditional OE approaches produces persistent residuals well above the noise levels due to the strong constraint. For the SVD method, the $X_{CO2}$ column mean (first principal component in the **z**-basis) is nevertheless bias free. However, the OE method shows a bias when the results are projected to the **z**-basis. (right) Results projected to the **x**-basis also show a clear bias for the OE method, though the $CO_2$ profile is well behaved. This shows that when using the Bayesian prior as a regularization to get a well behaved $CO_2$ profile, one runs the risk of overconstraining the retrieval and incurring a bias in the column mean. See Fig. 8 (right) for ensemble results.

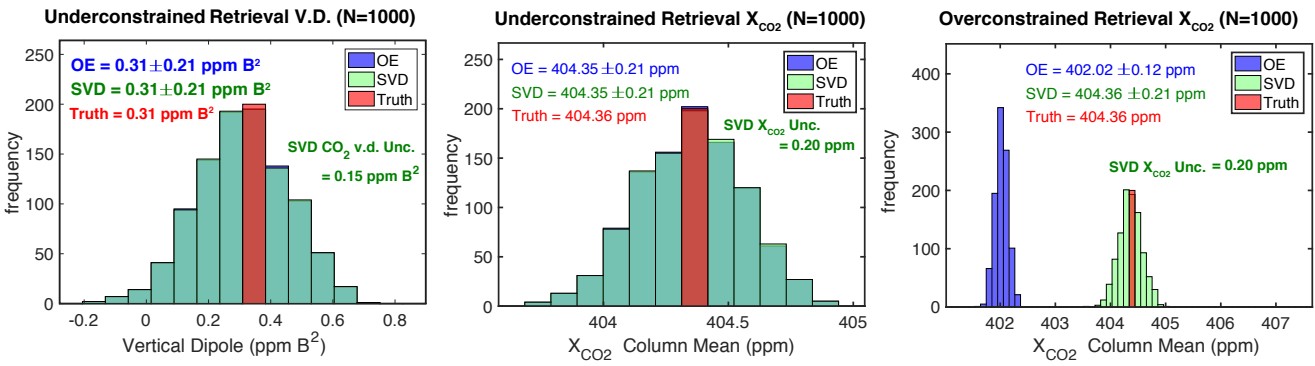

**Figure 8.** (Color online) Ensemble results for retrieved $CO_2$ parameters from numerical simulations. For a weak constraint (see Figure 6), the SVD method and OE methods both produce good results in the **z**-basis for both the $CO_2$ vertical dipole moment (left) and $X_{CO2}$ column mean (center), which constitute the first two principal components. Results are in line with the expected variance, $\mathbf{S_z}$ from the SVD method. Under strong constraint (right, Figure 7), the OE method produces a smaller standard deviation, but starts to incur a bias, whereas the SVD method continues to produce accurate results, but with no reduction in the variance. Note - For the strong constraint case, the SVD $CO_2$ vertical dipole moment is not retrieved.



the prior uncertainty. This uncertainty is specified in the prior covariance matrix $\mathbf{S}_a$. In our simulations, the prior uncertainty of the $CO_2$ concentration at each level ($x_i$ in $\mathbf{x}$) is varied between 0.1-100% (strong-weak) depending on the case.

SVD retrievals are constrained by the number of principal components used in the line-fitting. While the constraint is applied in qualitatively different ways to the two retrieval methodologies, the effect is somewhat similar particularly for weak

constraints, since the SVD method is the limiting case of a weak prior constraint (discussed in section 3.3). For the SVD method, we retrieve between 1-4 (strong-weak) $CO_2$ principal components depending on the case.

### 5.3   Case 1: Underconstrained fit

In this case, we set the prior uncertainty in $\mathbf{S}_a$ for the traditional OE method to be 100%. For the SVD retrieval, we include four $CO_2$ principal components (see Figure 3 for a description of the components) in the fit. The results for a single simulated

measurement are shown in Figure 6 (this can be contrasted with Fig. 7, which has results for an overconstrained fit). As expected, the OE retrieval (and SVD retrieval projected to $\mathbf{x}$-basis) results in a $CO_2$ column with widely varying mixing ratios. Nevertheless, in the SVD $\mathbf{z}$-basis, both methods produce meaningful column averaged $X_{CO2}$ results. This is due to the orthogonality of the principal component basis, ensuring that lower order components are unaffected by large swings or errors in higher order components.

Ensemble results (Figure 8, left and center) further confirm that the SVD and OE methods both produce bias-free results in the principal component $\mathbf{z}$-basis. In addition, we see that the calculated uncertainty from (26) is in good agreement with the variance in the SVD ensemble as well as the weakly constrained OE ensemble.

### 5.4   Case 2: Overconstrained fit

A strong constraint puts restrictions on the state vector and prevents a retrieval from fully minimizing the residual. Here, we

set the prior uncertainty in $\mathbf{S}_a$ for the traditional OE method to be 0.1%. For the SVD retrieval, we allow just 1 $CO_2$ principal component in the fit. The effects of a strong constraint in each case is shown in Figure 7. While the OE method shows a clear bias towards the prior mean ($\mathbf{X}_a$), the SVD method is still able to retrieve an accurate $X_{CO2}$ column mean. Again, this is due to the orthogonality of the principal component basis, ensuring that lower order components are unaffected by the absence of higher order components in the fitting.

Ensemble results (Figure 8, right) further illustrate the bias in the traditional OE method with strong weighting towards the prior mean. Correspondingly, with a low uncertainty in the prior mean, the traditional OE retrievals produce a lower variance. Thus in order to benefit from the availability of prior information, the prior mean needs to be in good agreement with the true mean. Ensemble results for the SVD method show that it remains bias free. The calculated uncertainty for the SVD method, which is independent of the number of principal components, is unchanged, and ensemble results confirm the same.

Although a rather extreme constraint has been applied for the traditional OE method, the results show that there are intrinsic problems in using a constraint that is too strong. Often, such biases are subtle and less obvious, but nevertheless affect flux measurements, which are based on several thousand soundings and sensitive to small biases. In contrast, the SVD approach is more robust.




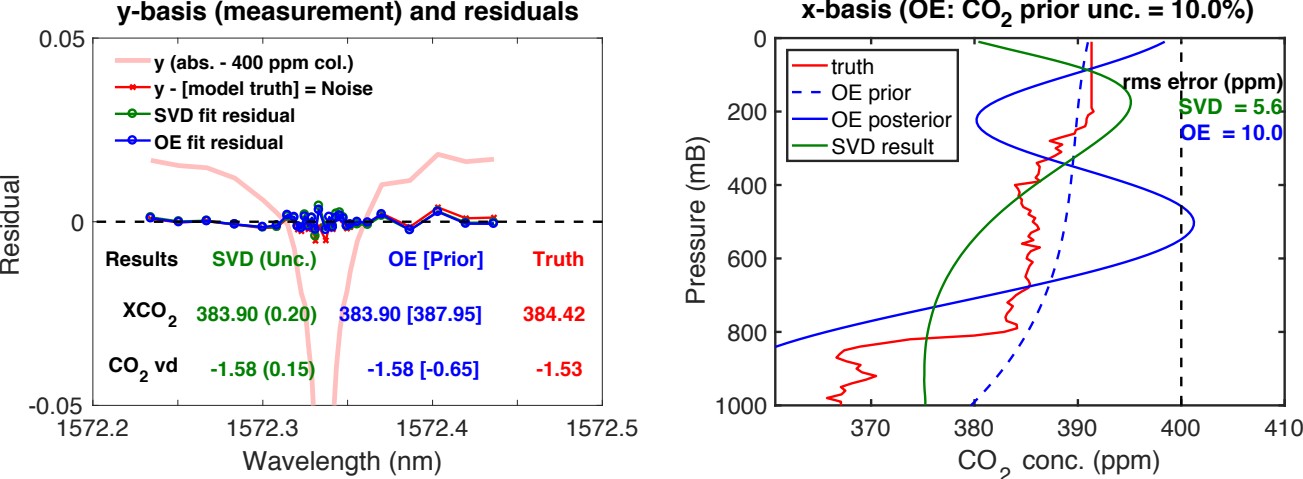

**Figure 9.** (Color online) Sample simulated measurement of vertical dipole moment using the SVD method and appropriate constraint - (left) The SVD and OE approaches demonstrate good convergence, and since the SNR is relatively high, the residuals are small. Results projected to the **z**-basis show reasonable performance of both techniques for retrieving the $X_{CO2}$ column mean (first principal component) and vertical dipole moment, with agreement within the expected variance. (right) Both methods detect the overall decrease in the $CO_2$ concentration at low altitudes. Despite a helpful Bayesian prior, the OE retrieval still differs significantly from the true profile. In addition, despite the $CO_2$ profile differing significantly from the uninformative prior used in the SVD method, the bias in the retrieved $X_{CO2}$ is small and for this instance, likely due to random error (see Fig. 10 for more precise comparisons using an ensemble of simulations).

## 5.5 Case 3: Extracting vertical $CO_2$ information using the vertical dipole moment term

Having demonstrated the SVD method's general robustness, we now look at the extraction of vertical information about the $CO_2$ distribution. During a flight over Iowa during the summer crop season in 2011, *in situ* measurements of the atmospheric $CO_2$ concentration profile showed a sharp 15 ppm drawdown in the boundary layer compared to the free troposphere (Ra-
5   manathan et al., 2015). When projected on the basis of principal components, this corresponded to a significant vertical dipole moment of -1.53 ppm $B^2$. Figure 9 shows the SVD method capture the vertical dipole moment with an uncertainty of ±0.15 ppm $B^2$. When projected back to the **x**-basis, the $CO_2$ vertical profile reconstructed from the principal components shows that the SVD method is able to reproduce the overall shape, but not the sharp increase in the planetary boundary layer. The OE method produces similar results despite a helpful prior from climatology data being used. Biases in the **x**-basis are still rather
10   high, >5 ppm.

    Figure 10 highlights the performance of the SVD and traditional OE methods in measuring the column mean. For the SVD method, we look at ensemble results for several choices in the number of principal components, ranging from 4 (undercon-strained) to 1 (overconstrained). For the OE method, we correspondingly vary the prior mean uncertainty (for each layer) from 100% (underconstrained) to 0.1% (overconstrained). We look at the variance and the bias of the $X_{CO2}$ column mean. At the




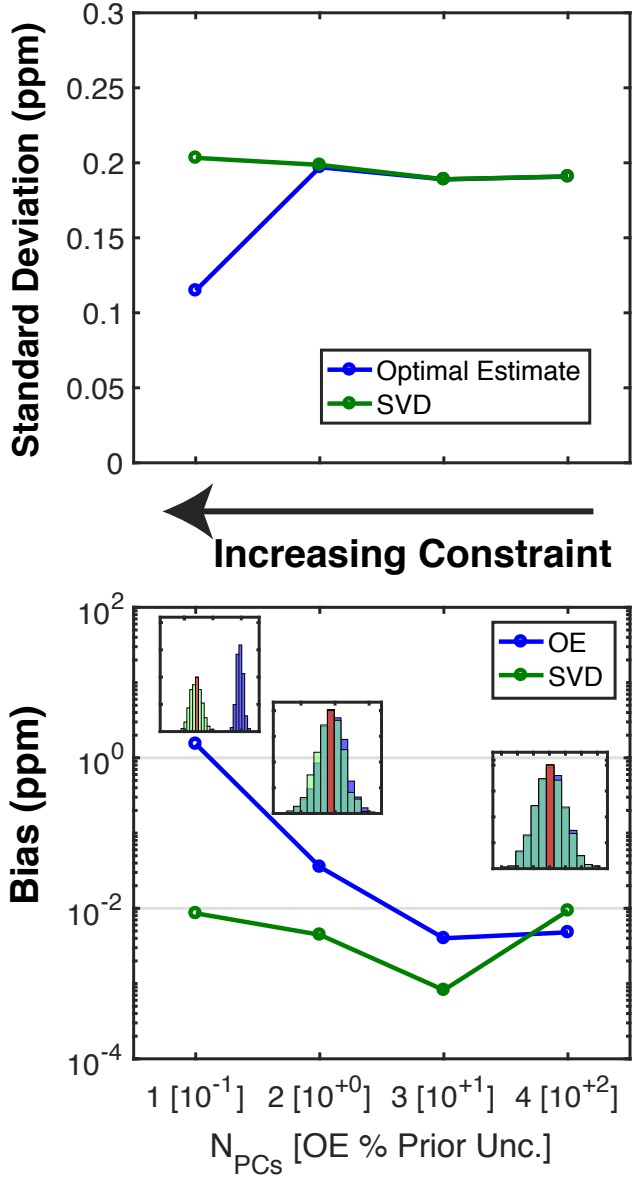

**Figure 10.** (Color online) Robust measurement of the $X_{CO2}$ column mean by the SVD method - Results from ensembles of 1000 numerical experiments show that for retrievals using the SVD method for a range of constraints (one to four principal components), the variance (top) and bias (bottom) in the column mean $X_{CO2}$ are robust. In contrast, a similar change in constraint when using the OE retrievals (changing prior uncertainty in the layer $CO_2$ mixing ratio from 100% to 0.1%), produces a decrease in the variance of $X_{CO2}$, but also a sharp increase in bias, above the 0.5 ppm accuracy needed for reasonable $CO_2$ flux inversions. Insets in the lower plot illustrate the ensemble distributions at different constraint points as done in Figure 8 (center and right plots).





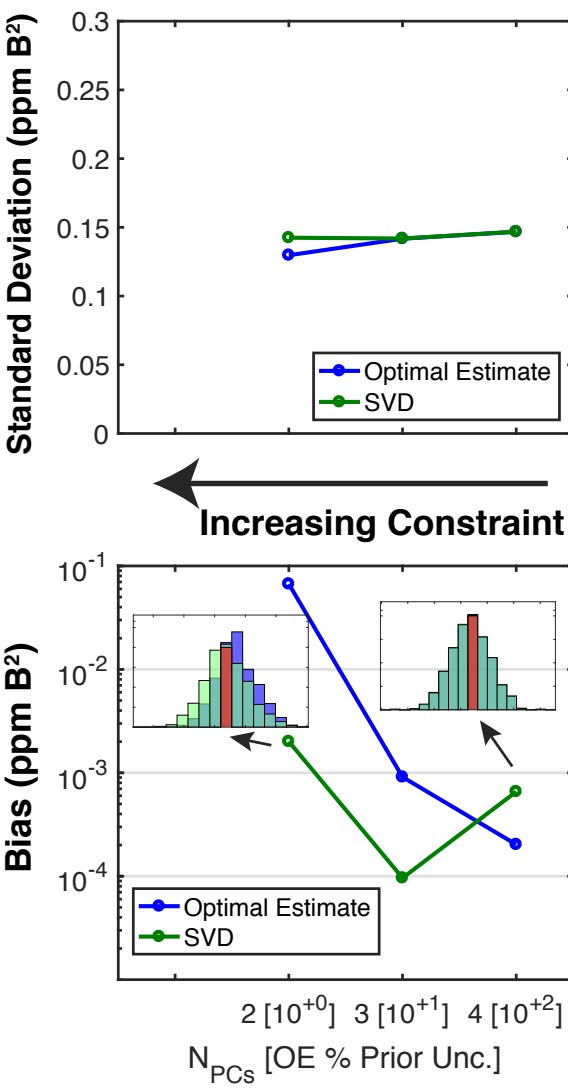

**Figure 11.** (Color online) Robust measurement of the $CO_2$ vertical dipole moment by the SVD method - As in figure 10, we look at the variance (top) and bias (bottom) in the retrieved $CO_2$ vertical dipole moment for ensembles of 1000 numerical experiments at varying constraints. As with the column $X_{CO2}$, the SVD results are robust and unaffected by the changing number (two to four) of principal components. In contrast, the OE results incur a significant bias (0.1 ppm B$^2$) when the prior uncertainty (the regularizing constraint) for the $CO_2$ mixing ratio at each layer is set at 1%. Insets in the lower plot illustrate the ensemble distributions at the minimum and maximum constraint as done in Figure 8 (left plot).



weakest constraints, the SVD and OE methods behave similarly as expected. As the level of constraint is increased, the OE measurement starts to have a lower variance, but incurs a bias since the assumed prior $CO_2$ profile differs from the truth (see figure 9). The SVD method column mean, in contrast is unaffected, despite the uninformative prior (400 ppm uniform column) also differing significantly from the truth. This illustrates the robustness of the SVD method.

Figure 11 shows similar behavior for the retrieved vertical dipole moment. As with the column mean, the SVD and traditional OE methods behave similarly at weak constraints. As the constraint is increased the OE method starts to have a lower variance, but also incurs a significant bias.

## 6  Discussion

The SVD framework and its use of principal components provides the mathematical basis to determine what information can
be extracted from GHG column absorption measurements. Section 3.5 confirms the notion that the retrieval of a column mean using least-squares line fitting of an absorption spectrum yields a bias-free estimate of the $X_{GHG}$, regardless of the shape of the profile used in the prior (which turns out to be uninformative). Beyond the retrieval of the column mean, the SVD framework identifies higher order modes such as the vertical gradient (vertical dipole moment) that can potentially be retrieved with sufficient measurement precision.

Although the numerical results from the SVD method have been shown for the $CO_2$ Sounder lidar instrument, the SVD method itself can also be applied to total column absorption measurements from ground-based and satellite spectrometers since those instruments also measure pressure-broadened absorption lineshapes in the atmosphere. A key parameter affecting the principal components and the precision to which they can be retrieved is the instrument spectral resolution or linewidth (see Figure 4). While ground-based spectrometers like TCCON and mini-LHR have a high spectral resolution and can retrieve
more than one principal component, others such as the lower resolution Bruker EM27 (0.5 cm$^{-1}$ resolution, Gisi et al. (2012)) will have significantly poorer precision for higher order principal components. Furthermore, satellite instruments like GOSAT and OCO-2, besides having coarser spectral resolution, have the additional complication of aerosol scattering mixed with the signal (due to the lack of range gating of the surface reflected signal), which can limit the accuracy of the retrieved principal components.

### 6.1  Advantages of using Principal Components

The primary benefits of using the SVD method with retrievals in the principal component basis can be summarized as follows:

1. Retrieval of higher order terms of the greenhouse gas vertical distribution (beyond the column mean) in the atmosphere

2. No bias from the use of an uninformative prior

3. Orthogonality of principal components leading to robust retrievals independent of the degree of constraint (number of
components solved for)





The robustness of the SVD method makes it useful in situations where the prior state is not well known or the uncertainty in the prior is not well quantified. For instance, $CO_2$ vertical profiles are measured only at a few locations around the Earth. While $CO_2$ retrievals over those select locations could benefit from the use of a Bayesian prior, retrievals over remote regions far from those places would better be served by the SVD method since the prior knowledge of the $CO_2$ profile is not well known (see

section 6.4 for when to choose SVD over OE). This is a key virtue of the SVD method.

The robustness of the SVD method also may makes it easier to use in an operational environment, where atmospheric and surface conditions can change the measurement precision significantly. Rather than using advanced retrieval methods to get vertical information separate from the main retrieval of the column mean (as in Kulawik et al. (2016)), one can simply retrieve several principal components in the main retrieval itself (and keep them as part of the main product), but only assimilate the

components that have sufficient precision into GHG flux models.

Furthermore, in performing the retrieval in the principal component basis, the SVD method requires fewer computations than the OE method, which works in the full model basis. This has the potential to make the retrieval faster and more efficient. In addition, the reduced basis of mutually orthogonal principal components makes retrieval analysis easier. Troubleshooting systematic or forward model errors are also simpler in the principal component basis since the basis is smaller and the prior is

uninformative, allowing one to more easily see the effects (manifested as a bias) on the different components.

### 6.2   Practical application of the SVD method to GHG retrievals

In practice, interference from other gas species in the atmosphere (for instance, water vapor) and instrument systematic errors prevent the simultaneous realization of all benefits listed in section 6.1 with the use of the SVD method. Nevertheless, one can use the SVD framework to analyze the problem and try to get most of the benefits. While a full analysis of all interferences

and systematic errors and their effects on SVD retrievals is beyond the scope of this work, we give a simple example to show how certain types of interferences can be treated within the SVD framework.

The presence of a water vapor line at the shoulder of the $CO_2$ absorption line described in this work (also see Abshire et al. (2017)) causes the principal components to have combinations of water vapor and $CO_2$ mixing ratios that are not physically meaningful. If one chooses to use the principal component retrieval basis, one gets benefits 2 and 3 described above, but not

benefit 1. If one chooses to keep the $CO_2$ mixing ratio principal components separate from the water vapor components in the retrieval basis, one gets benefits 1 and perhaps benefit 2, but not benefit 3. One can also use techniques like clumped fitting (Abshire et al., 2017) to use information based on spatial correlations of the water vapor mixing ratio as well as other systematic effects to try and get at all three key benefits.

### 6.3   Comparing the SVD method to Kulawik et al. (2016)

The SVD method discussed here bears some similarity to the approach used by Kulawik et al. (2016) to extract vertical information. Both methods use an uninformative prior and retrieve two pieces of information about the $CO_2$ column. Although the LMT (lowermost troposphere) $CO_2$ product is easier to relate to given that it represents the mixing ratio of the bottom 2.5





km of the atmosphere, it does have some sensitivity (with opposite sign) of higher altitude $CO_2$ concentrations similar to that of the $CO_2$ vertical dipole moment term discussed in this work.

There are also some important differences between the two methods. In using principal components, the SVD retrieval produces orthogonal parameters that have uncorrelated errors and thus errors in the $X_{CO2}$ are uncorrelated with those of the

$CO_2$ vertical dipole moment. In contrast, given the way the information is partitioned in Kulawik et al. (2016), the LMT product is expected to be negatively correlated with the U (upper atmosphere) product. In addition, it is expected to have higher precision than the SVD vertical dipole moment for the same data. Future work will involve a quantitative comparison of retrievals using the two techniques on the same absorption data, which could better illustrate the advantages of each of these methods.

## 6.4   Implications of using the traditional OE method

Going beyond the domain of trace gas retrievals to the broader problem of atmospheric sounding, the simulations shown in this paper underscore the importance of choosing a proper Bayesian prior and prior covariance if using the OE method. Ideally, the choice of these parameters will be from a large sampling of the true state space. In the absence of such data, the prior mean may be different from the true mean. Setting or tuning of the constraint from the Bayesian prior for the purpose of regularization of

the retrieval problem runs the risk of overstating prior knowledge and thus causing a bias.

In choosing between the SVD method and the traditional OE method, one needs to factor in the quality of the prior information (See Figure 12) relative to the signal to noise ratio of the measurement. While the SVD method is always the safer option (less susceptible to bias), in situations when the measurement is noisy but the Bayesian prior is well characterized, the OE retrieval will result in a lower variance.

## 7   Conclusions and future work

We have described an approach to deducing vertical information from column GHG retrievals based on the Singular Value Decomposition. The SVD approach does not require an assumption of a prior distribution of the GHG profile for regularizing the retrieval problem, and by using the principal component basis for retrievals, the prior is rendered as uninformative. Simulations comparing the SVD method to the traditional Bayesian OE (using an informative prior) show that the SVD method

is more robust and better suited to situations where prior knowledge of the $CO_2$ concentration and distribution is lacking or poorly characterized.

In this work, we have assumed a perfect forward model and only random errors in the measurement. This is necessary first step check for the feasibility of the method. However, in practice, other sources of error such as imperfect instrument calibration, imperfect knowledge of atmospheric state and forward model approximations play important roles. Our preliminary

attempts using $CO_2$ Sounder data from airborne field campaigns have shown that small errors in spectroscopy arising from neglecting the non-Voigt component of the lineshape can cause significant biases. These errors are beyond the scope of this paper and will be addressed in future work.





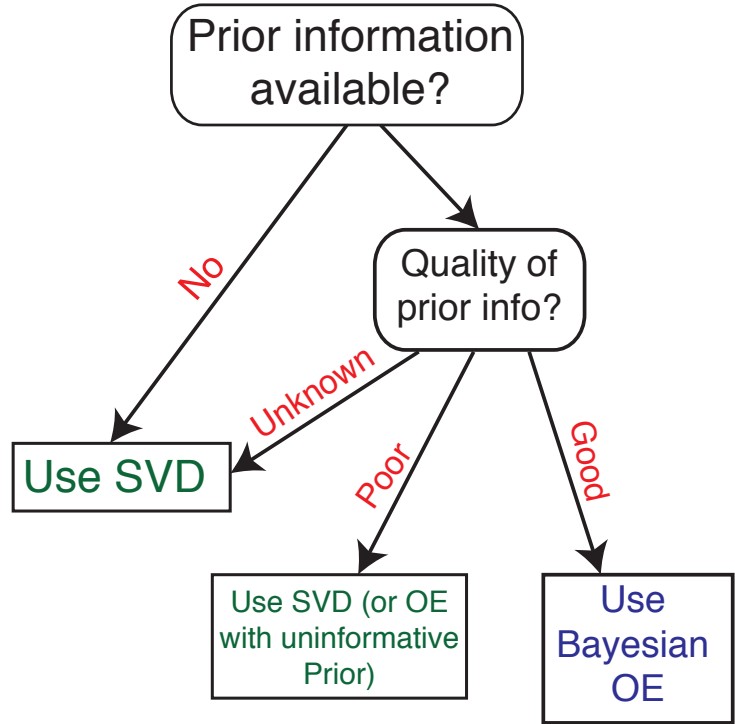

**Figure 12.** Decision tree for suitable retrieval approach - The quality of prior information compared to the signal to noise ratio (SNR) determines which retrieval method would be better suited. The SVD method (with principal components retrieved) is robust and can be applied to a range of situations. However, in situations where the prior information is good (relative to the measurement SNR), the OE method offers a clear advantage of a lower variance in the retrieved $X_{CO2}$.

Another interesting topic is extending this work to non-linear forward models, where the minimization of the loss function in Eqn. (6) amounts to solving a non-linear least square problem. In the traditional OE framework, the *maximum a posteriori* solutions are popularly solved using some variation of Newton's method. Since we have shown that the SVD method can be viewed as an OE algorithm, its extension to the non-linear forward model can similarly make use of the iterative Newton's

5  method (Rodgers, 2000) to solve for the *maximum a posteriori* solutions. Preliminary numerical simulations indicate that the SVD method is still unbiased for non-linear forward models as long as $\mathbf{F}(\cdot)$ is sufficiently "smooth" at the retrieved state, though further studies are required.

Future work will also explore other aspects of the measurement problem such as determining the optimal wavelength sampling. In contrast to passive spectrometers, which can have a large number of samples, lidar instruments bear some cost for

10  each additional sample. While in theory one needs a wavelength sample for each principal component retrieved, in practice one needs to oversample the line to help reduce systematic errors (control biases). Determining the optimal wavelength sampling,



to best obtain information about the vertical distribution of the GHG while keeping biases low, is important in the design of space-based IPDA lidar instruments for GHG measurements.

*Acknowledgements.* AKR, XS, JM and JBA are grateful for support from the NASA ASCENDS Mission Science definition activity.





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
