# Peer review of "A Singular Value Decomposition framework for retrievals with vertical distribution information from greenhouse gas column absorption spectroscopy measurements"

_Atmospheric Measurement Techniques, 2018_

## Referee Comment (RC1) · Anonymous Referee #1 · 1 Mar 2018

General comments:

This is an important and well executed paper which should be published without delay. The subject of greenhouse gas retrievals from remote measurements is a critical one for advancing scientific knowledge of climate processes using an extensive new generation of instruments, networks, and techniques including OCO-2, GOSAT, TCCON, and IPDA lidar measurements.

The authors clearly develop the mathematics of the SVD technique and present a well-

chosen example of its application to GHG retrievals. While the SVD technique has a long history, the authors' careful analysis of it's mathematical relationship to the 'de facto standard' OE, and its practical comparison to OE, in the context of GHG retrievals, is new and very informative.

As one of the principal developers of the use of OE for GHG retrievals from remote spectral measurements, I do have reservations about the practical use of SVD in place of OE. Namely the main error sources in such retrievals are forward model errors and interference from state vector elements of less interest than the target species. Also, the forward models for the infrared spectral measurements exemplified by OCO-2 etc. are moderately non-linear. Neither of these fundamental issues are addressed, as the authors themselves note in section 7, calling for future work. However the current paper makes it clear that further investigation and development of SVD in this context may lead to important advances.

Specific comments:

P 4 line 22: 'better representing' I don't know what is meant here, please restate.

P 7: forward model definition. I find this section confusing. The measurement vector y is described as the deviation in the absorption from that corresponding to xu, but that is not a measurable quantity and the statement is contradicted by Eq. (4). Near the bottom of page 8 it is claimed a valid choice for the reference profile is xu = 0, so that y is a 'deviation' from zero.

I believe this is correct in the end, and exploits the assumed linearity of the problem, but it is still not entirely clear to me, and I think the concepts should be better explained.

P 21 line 7: 'to the create'

P 28 line 11: 'bias-free estimate' This is not true in general, as the authors have themselves noted on P 4 line 24. It should be qualified or its applicability defined.

---

## Referee Comment (RC2) · Anonymous Referee #2 · 21 Mar 2018

Review of "A Singular Value Decomposition framework for retrievals with vertical distribution information from greenhouse gas column absorption spectroscopy measurements" by Ramanathan et al.

Overall comments:

Development of a SVD approach for CO2 lidar instruments is useful and worth publishing. However there are some major issues in this paper that need to be addressed before it can be fully reviewed. Partitioning the profile into preset shapes is discussed

in Tukiainen et al. (2016) for TCCON CH4, although the retrieved shapes were based on prior covariance. The error analysis in this paper should be compared to Tukiainen (2016).

One of the main ways this approach is validated is by comparisons to optimal estimation, however the optimal estimation retrievals do not look comparable to profile retrievals from OCO-2 available in the L2 standard products. For example Figure 6 shows 5 oscillations in the retrieval on the order of 50 ppm. OCO-2 retrieved profiles do not show these types of oscillations. It appears from p. 24 line 1-2 that the constraint used in OE is diagonal. The constraint used should match O'Dell, 2012 (Figure 2) which has strong off-diagonal correlations. Comparing the SVD retrievals to state of the art OE retrievals will be useful.

The second issue in this paper are the claims in the abstract that SVD results in unbiased results and is therefore better than OE. While it is true that the basis functions may not need to be constrained if truncated at whole degrees of freedom, and there may be no biases in the mapped space, the translation of the basis functions into a profile can result in biases and these should be quantified. The biases introduced by this approach should estimated by calculating the linear estimate for different true states, e.g. Tukiainen (2006) Fig. 3 shows the difference between AirCore and smoothed AirCore for methane and a similar SVD approach. The column difference between AirCore and smoothed AirCore (or some other set of trues) would give the bias and error resulting from the SVD mapping using Eq. 34 from this paper. Section 3.3 is also hard to follow.

The authors should clarify how many basis functions are selected. If there are 1.6 degrees of freedom, are 2 basis functions used? If 2, won't the retrieval need some constraint? If 1, won't that throw away information? Kulawik et al. (2017) used 1.6 degrees of freedom from GOSAT to get 2 parameters each with about 0.8 degrees of freedom (so some a priori component in the retrieved values). Would this approach be able to get any vertical information with 1.6 degrees of freedom?

[Figure]

I do not follow Figure 4. What are the units on the y-axis? Are the authors aware that OCO-2 has better precision and more degrees of freedom than GOSAT? This figure suggests the opposite.

I look forward to reviewing this paper when the authors improve the OE results and more carefully characterize the bias and errors compared to the current OE method.

———————————————————

---

## Author Comment (AC1) · 18 Apr 2018

**Response to Anonymous Referee 1**

- **P 4 line 22: "better representing" I don't know what is meant here, please restate.**

  Changed the sentence for more clarity from

  "Such simple methods have the advantage of better representing the instrument

measurement, and enabling more feedback on instrument performance."

to

"Such simple methods have the advantage of enabling more feedback on instrument performance by virtue of forcing the retrieval to derive certain information strictly from the measurement even when non-optimal."

- **P 7: forward model definition. I find this section confusing. The measurement vector $y$ is described as the deviation in the absorption from that corresponding to $x_u$, but that is not a measurable quantity and the statement is contradicted by Eq. (4). Near the bottom of page 8 it is claimed a valid choice for the reference profile is $x_u$ = 0, so that $y$ is a 'deviation' from zero. I believe this is correct in the end, and exploits the assumed linearity of the problem, but it is still not entirely clear to me, and I think the concepts should be better explained.**

We agree with the reviewer. We have modified Eqn. 4 to include the noise term $\mathrm{epsilon}$

The point of the equations at the bottom of page 8 were to use the offset and scale choices that we have at our disposal in setting how $x$ relates to the $CO_2$ profile (in units of ppm). Our specific choice was made based on making the SVD equations least complicated. In the revised version, we will include an additional equation to explicitly show that relationship and include a sample calculation for clarity.

In the examples we show, we set the uninformative prior to be a 400 ppm uniform column. With $x_u$ being zero, $\hat{x}$ having all elements 0 corresponds to a uniform column of 400 ppm. An element of $\hat{x}$ having a value of 0.02 corresponds to that layer in the atmosphere having a mixing ratio of $(1 + 0.02) \times 400 = 408$ ppm. Similarly, an element of $\hat{x}$ having a value of -0.02 corresponds to that layer in the atmosphere having a mixing ratio of 392 ppm.

The element in $\mathbf{x_u}$ corresponding to the surface reflectance or signal level ($x_0$) also has degrees of freedom for the offset and scale. Thus, one can set $\mathbf{x_u}$ to be zero with no loss in generality.

We have revised the text at the bottom of page 8 to

"In the above equations, we have carefully exercised our choice in linearly mapping the physical world to $\mathbf{x}$ by setting

$$\mathbf{x}_u = 0$$

for simplicity, and scaling $\mathbf{x}$ such that $x_i = -1$ corresponds to the GHG concentration of the $i^{\text{th}}$ layer in the atmosphere being zero. As per Eqn. 3, $\mathbf{F}(\mathbf{x}_u)$ is a constant, which can also be set to zero with no loss in generality. These sorts of transformations are fairly standard in the literature and make the equations less complicated. "

We have also expanded our description of the numerical simulation methodology pertaining to the SVD method:

"For the SVD approach, we set the uninformative prior $\mathbf{x}_u$ to be a uniform 400 ppm $CO_2$ profile and anchor our definition of $\mathbf{x}$ to it. From this, $x = -0.02, 0, 0.02$ would correspond to mixing ratios of 392, 400 and 408 ppm respectively. "

- **P 21 line 7: "to the create"**

We have corrected this to "to create"

- **P 28 line 11: "bias-free estimate" This is not true in general, as the authors have themselves noted on P 4 line 24. It should be qualified or its applicability defined**

We have changed the sentence from

"...confirms the notion that the retrieval of a column mean using least-squares line fitting of an absorption spectrum yields a bias-free estimate of the $X_{\text{GHG}}$,

regardless of the shape of the profile used in the prior (which turns out to be uninformative)"

to

"...confirms the notion that the retrieval of a column mean using least-squares line fitting of an absorption spectrum yields an estimate of the $X_{GHG}$ without incurring bias from the regularization or retrieval, regardless of the shape of the profile used in the prior (which turns out to be uninformative) "

---

## Author Comment (AC2) · 18 Apr 2018

**Response to Anonymous Referee 2**

- **Partitioning the profile into preset shapes is discussed in Tukiainen et al. (2016) for TCCON CH4, although the retrieved shapes were based on prior covariance. The error analysis in this paper should be compared to Tukiainen (2016).**

   Although we agree with the referee that a comparison of the error analysis with

that of Tukiainen et al. (2016) would be useful, we feel it would be beyond the scope of this work. While there are some similarities between the methods, namely that principal components with truncation are used to solve the problem, there are two key differences. The first, as the referee has noted, is that Tukiainen used the prior covariance to determine the terms in the dimension reduction. The second key difference is that we report results in the principal component basis ($z$) rather than full model space ($x$) for the purpose of either GHG flux modeling and retrieval validation.

The SVD method as described in our work has the key advantage of being able to retrieve scientifically useful quantities with any bias from the regularization process even in the absence of prior information about the GHG profile. Projecting such results onto the full model space will require information from the prior and thus introduce bias, since the prior is meant to be uninformative.

Nevertheless, it will be useful to make a one-to-one retrieval and error analysis comparison on a TCCON like system between our use of the SVD method and that of Tukiainen et al (2016). While that is not within the scope of this work, we will definitely consider it for future work.

- **One of the main ways this approach is validated is by comparisons to optimal estimation, however the optimal estimation retrievals do not look comparable to profile retrievals from OCO-2 available in the L2 standard products. For example Figure 6 shows 5 oscillations in the retrieval on the order of 50 ppm. OCO-2 retrieved profiles do not show these types of oscillations. It appears from p. 24 line 1-2 that the constraint used in OE is diagonal. The constraint used should match O'Dell, 2012 (Figure 2) which has strong off-diagonal correlations. Comparing the SVD retrievals to state of the art OE retrievals will be useful**

We agree that a comparison of the SVD retrievals to the state of the art OE retrievals will be useful. However, there are several complications that go into the

choice of a Bayesian prior such as that used in OCO-2, such as local meteorology, vertical mixing and confidence in global GHG models at the location in question. Our intent behind this work was to showcase the SVD method and compare and contrast it with the OE method using a simplified system. Hence, we decided to use a conservative, 200 mB $1/e^2$ vertical correlation distance in the $CO_2$ mixing ratio in the atmosphere for off-diagonal terms (Page 22, line 9).

We do plan future work to make a comparison between the state-of-the-art OCO-2 retrievals and one based on the SVD method.

In the paper, we have expanded on the description of the Bayesian prior chosen: "For the OE approach, a proper choice of a Bayesian prior would factor in local meteorology, vertical mixing and confidence in global GHG models at the location in question. However, for the purpose of illustration of the workings of the OE method, we have kept the Bayesian prior mean and covariance simple. The Bayesian prior mean and variance (diagonal terms on the covariance matrix) are chosen on a case-by-case basis. For the prior covariance (off diagonal terms in the covariance matrix), we assume a 200 mB $1/e^2$ vertical correlation distance in the $CO_2$ concentration in the atmosphere. "

- **The second issue in this paper are the claims in the abstract that SVD results in unbiased results and is therefore better than OE.**

It was not our objective to show that SVD is "better than OE". In Section 3.3, we showed that SVD is equivalent to an OE estimator with an uninformative prior and a Moore-Penrose pseudoinverse. That is, SVD can be considered as a subclass within the OE framework. Also, as seen in Figure 12, we specifically advocate the OE method when good quality prior information is available since it gives the best estimate.

In the paper, we drew a distinction between the types of priors used within OE (informative) and within SVD (uninformative), and we derived some interesting

properties of the two choices of priors. They each have their own strength, which we summarize in a new paragraph in the Summary section:

"Intuitively, OE derives an estimate of the state using both the measurement and prior knowledge, while SVD only uses just the measurement to inform its estimate. When the prior information is correct, there is no doubt that OE will have lower posterior uncertainty since OE can leverage an extra source of information to more efficiently derive its estimate. However, this efficiency comes at a potential cost when the prior is *incorrect*. For instance, we showed that when OE uses an incorrect prior mean, then the estimate is guaranteed to be biased. Estimates from the SVD method in the principal component basis, on the other hand, are insensitive to incorrect information coming from the prior. The choice between SVD and OE then mostly comes down to how well one understands the prior distribution of the state of interest."

- **While it is true that the basis functions may not need to be constrained if truncated at whole degrees of freedom, and there may be no biases in the mapped space, the translation of the basis functions into a profile can result in biases and these should be quantified.**

  The translation of the basis functions into a profile can indeed result in biases. We did show analytic expression for the bias of OE and SVD *in the original profile space* in the paper. They are in equation (30) and (33). For your convenience, we include them in this response. Suppose that OE and SVD both uses a wrong prior mean $_b$, which is different from the true prior mean $_a$, then the expected bias for OE is

$$Bias_{OE} = (-(_a^{-1} + _\epsilon^{-1\prime})_\epsilon^{-1-1\prime})(_b - _a). \tag{1}$$

And the SVD bias is

$$Bias_{SVD} = (-(_{\epsilon}^{-1\prime})_{\epsilon}^{+-1\prime})(_b-_a). \qquad (2)$$

Note that when OE uses the correct prior mean ($_b =_a$), then $Bias_{OE} = 0$. Also, when ($_{\epsilon}^{-1\prime}$) is invertible, then $Bias_{SVD} = 0$ regardless of the choice of $_b$.

Detailed error analysis with simulations was done in the z-basis because the full GHG profile (x-basis) is often not needed for use in GHG flux modeling. The retrieved parameters obtained from the z basis retrievals can often be directly mapped to a GHG column mean and other higher order components. It should be noted that this is the basis of line-fitting methods even if they don't explicitly use the SVD method, since they derive $X_{\mathrm{GHG}}$ strictly from the measurement, whose information is contained within the SVD basis. $X_{\mathrm{GHG}}$ is ingested or assimilated into GHG flux models today. Higher order components like the vertical dipole moment can also be similarly ingested based on their information content as has been described in Joiner and Da Silva, *"Efficient methods to assimilate remotely sensed data based on information content"* (1998). A reference to Joiner and Da Silva (1998) has been added. A sentence has been added to the end of the last paragraph in section 2.1 "Joiner and Da Silva (1998) describe a method that can ingest such components into an assimilation model based on their information content. "

- **The biases introduced by this approach should estimated by calculating the linear estimate for different true states, e.g. Tukiainen (2006) Fig. 3 shows the difference between AirCore and smoothed AirCore for methane and a similar SVD approach. The column difference between AirCore and smoothed AirCore (or some other set of trues) would give the bias and error resulting from the SVD mapping using Eq. 34 from this paper.**

The intent of the SVD approach described is for functional retrievals that can provide inputs into GHG flux models either in the absence of prior GHG profile

information or when such information is of unknown quality or potentially biased. As stated above, these do not require a retrieval of a full vertical profile. Nevertheless, we have shown the expression for calculating the error in the GHG profile. Please see Eqn (1) and (2) from the response above.

- **Section 3.3 is also hard to follow.**

  We rewrote section 3.3 slightly to indicate that the OE under certain assumptions is equivalent to SVD . Specifically, we rewrote the equation block in (22) to indicate that the retrieved value $_{uOE}$ arising from OE with an uninformative prior and pseudo-inverse is *identical* to the retrieved value $_{SVD}$ arising from SVD.

- **The authors should clarify how many basis functions are selected. If there are 1.6 degrees of freedom, are 2 basis functions used? If 2, won't the retrieval need some constraint? If 1, won't that throw away information? Kulawik et al. (2017) used 1.6 degrees of freedom from GOSAT to get 2 parameters each with about 0.8 degrees of freedom (so some a priori component in the retrieved values). Would this approach be able to get any vertical information with 1.6 degrees of freedom?**

  The calculation of degrees of freedom depends on the state of prior knowledge of the true state. In the SVD framework as we have used it, with prior information being absent, the retrieval process works the same whether the prior GHG profile is well known or poorly known. The key metric is the posterior uncertainty of the retrieved quantities. One can compare the posterior uncertainty against that of the prior knowledge of the system.

  With retrievals in the principal component basis, the retrieved parameters are orthogonal to each other. The practical implication of this is that the mean and variance of a retrieved parameter is independent of the inclusion or exclusion of higher order parameters. This is illustrated in figures 10 and 11.

If a higher order parameter is retrieved despite insufficient information, that parameter will have a posterior uncertainty that is too large to be useful. If 2 degrees of freedom are retrieved when the Bayesian framework projects 1.6 degrees of freedom, the second order component will have an uncertainty that is slightly too large to be useful on its own. As mentioned previously, it will not affect the retrieval otherwise.

- **I do not follow Figure 4. What are the units on the y-axis? Are the authors aware that OCO-2 has better precision and more degrees of freedom than GOSAT? This figure suggests the opposite.**

We are aware that OCO-2 has better prevision and more degrees of freedom than GOSAT. The purpose of Figure 4 was to illustrate the effect of spectral resolution, particularly on the higher order components. The overall precision depends on several factors including the light gathering capacity, detector sensitivity, integration time, etc.

We have replaced the words GOSAT and OCO-2 with regions indicating spectral resolutions of satellite instruments. Here is the new Figure 4 with caption

Caption to figure 4- Retrieval uncertainty versus instrument spectral linewidth for the first two $CO_2$ principal components (PCs) : While the column $X_{CO2}$ is largely unaffected by the spectral resolution, the precision of the $CO_2$ vertical dipole moment degrades strongly with poorer resolution. We assume a $CO_2$ instrument model, but with some instrument line broadening. The x-axis denotes the full-width at half maximum of the triangular instrument line shape used to broaden the $CO_2$ absorption. We assume photon shot noise with a SNR of 1000 for points with no $CO_2$ absorption. The spectral resolutions of TCCON, satellite GHG sensing spectrometers and the $CO_2$ Sounder instrument are indicated, though the calculations done in this work apply only to the $CO_2$ Sounder instrument.

**Retrieval Unc. for a $CO_2$ Sounder like instrument**

Figure showing X-axis: Instrument Lineshape width (cm$^{-1}$); left Y-axis: $X_{CO2}$ Uncertainty (ppm); right Y-axis: $CO_2$ V.D. unc. (ppm B$^2$). Legend: PC1: $X_{CO2}$ (ppm); PC2: $CO_2$ vertical dipole (ppm B$^2$). Labels: $CO_2$ Sounder, (TCCON), Satellite Instrument Spectral Resolution.

**Fig. 1.** Revised Figure 4

---

## Referee Report (RR1)

Review of the discussion paper "A singular Value Decomposition..." by A.K. Ramanathan et al.

As already stated by Reviewers 1 and 2, this paper tackles an important issue and I think that its topic fits perfevtly in AMT. In the following I will discuss some of the issues raised by the preceding reviews and will add a few comments of my own.

Rev. 1:
Comment:
P 28 line 11: "bias-free estimate" This is not true in general, as the authors have themselves noted on P 4 line 24. It should be qualified or its applicability defined

Reply:
We have changed the sentence from "...confirms the notion that the retrieval of a column mean using least-squares line fitting of an absorption spectrum yields a bias-free estimate of the XGHG, regardless of the shape of the profile used in the prior (which turns out to be uninformative)" to "...confirms the notion that the retrieval of a column mean using least-squares line fitting of an absorption spectrum yields an estimate of the XGHG without incurring bias from the regularization or retrieval, regardless of the shape of the profile used in the prior (which turns out to be uninformative)"

My view:
As stated below in more detail, this is not generally true but only if the signal measured is independent of any other altitude-dependent atmospheric state variable. Care should be taken to avoid any generalizing statement on bias-freeness but to limit the statements to particular results of the investigations made.

Rev. 2:
Comment:
Partitioning the profile into preset shapes is discussed in Tukiainen et al. (2016) for TCCON CH4, although the retrieved shapes were based on prior covariance. The error analysis in this paper should be compared to Tukiainen(2016).

Reply:
Although we agree with the referee that a comparison of the error analysis with that of Tukiainen et al. (2016) would be useful, we feel it would be beyond the scope of this work. While there are some similarities between the methods, namely that principal components with truncation are used to solve the problem, there are two key differences. The first, as the referee has noted, is that Tukiainen used the prior covariance to determine the terms in the dimension reduction. The second key difference is that we report results in the principal component basis (z) rather than full model space (x) for the purpose of either GHG flux modeling and retrieval validation. The SVD method as described in

our work has the key advantage of being able to retrieve scientifically useful quantities with any bias from the regularization process even in the absence of prior information about the GHG profile. Projecting such results onto the full model space will require information from the prior and thus introduce bias, since the prior is meant to be uninformative. Nevertheless, it will be useful to make a one-to-one retrieval and error analysis comparison on a TCCON like system between our use of the SVD method and that of Tukiainen et al (2016). While that is not within the scope of this work, we will definitely consider it for future work.

My view:
There are many retrieval methods which aim at minimizing the bias of the retrieval by avoiding informative prior and which keep the result stable basically by reducing the effective dimension of the result vector. Tukiainen (2016) is one of these, von Clarmann et al. (AMT 8, 2749-2757, 2015) would be another one, and I am sure there are a lot more. Comparison with each of these methods would be interesting but a line must be drawn somewhere and the decision which of these comparisons should be included in the paper should be left to the authors. The authors should not be forced to discuss any specific one of these papers unless a compelling reason is given that paper x is more important than paper y.

Comment:
One of the main ways this approach is validated is by comparisons to optimal estimation, however the optimal estimation retrievals do not look comparable to profile retrievals from OCO-2 available in the L2 standard products. For example Figure 6 shows 5 oscillations in the retrieval on the order of 50 ppm. OCO-2 retrieved profiles do not show these types of oscillations. It appears from p. 24 line 1-2 that the constraint used in OE is diagonal. The constraint used should match ODell, 2012 (Figure 2) which has strong off-diagonal correlations. Comparing the SVD retrievals to state of the art OE retrievals will be useful.

Reply:
We agree that a comparison of the SVD retrievals to the state of the art OE retrievals will be useful. However, there are several complications that go into the choice of a Bayesian prior such as that used in OCO-2, such as local meteorology, vertical mixing and confidence in global GHG models at the location in question. Our intent behind this work was to showcase the SVD method and compare and contrast it with the OE method using a simplified system. Hence, we decided to use a conservative, 200 mB 1/e2 vertical correlation distance in the $CO_2$ mixing ratio in the atmosphere for off-diagonal terms (Page 22, line 9). We do plan future work to make a comparison between the state-of-the-art OCO-2 retrievals and one based on the SVD method. In the paper, we have expanded on the description of the Bayesian prior chosen: "For the OE approach, a proper choice of a Bayesian prior would factor in local meteorology, vertical mixing and confidence in global GHG models at the location in question.

However, for the purpose of illustration of the workings of the OE method, we have kept the Bayesian prior mean and covariance simple. The Bayesian prior mean and variance (diagonal terms on the covariance matrix) are chosen on a case-by-case basis. For the prior covariance (off diagonal terms in the covariance matrix), we assume a 200 mB 1/e2 vertical correlation distance in the CO2 concentration in the atmosphere."

My view:
Again I agree with the authors. The purpose of the related section is validation, not competition. A common practice of validation is comparison with something well understood and easily traceable. The choice of a simplified OE approach appears to be rational and justifiable to me.

Comment:
The second issue in this paper are the claims in the abstract that SVD results in unbiased results and is therefore better than OE.

Reply:
It was not our objective to show that SVD is "better than OE". In Section 3.3, we showed that SVD is equivalent to an OE estimator with an uninformative prior and a Moore-Penrose pseudoinverse. That is, SVD can be considered as a subclass within the OE framework.

My view:
While I agree with the argument of the authors in its heart, I have some reservations with the use of the term "OE" in the context of ad-hoc priors or uninformative priors. See my own comments below.

Reply (cont'd):
Also, as seen in Figure 12, we specifically advocate the OE method when good quality prior information is available since it gives the best estimate. In the paper, we drew a distinction between the types of priors used within OE (informative) and within SVD (uninformative), and we derived some interesting properties of the two choices of priors. They each have their own strength, which we summarize in a new paragraph in the Summary section: "Intuitively, OE derives an estimate of the state using both the measurement and prior knowledge, while SVD only uses just the measurement to inform its estimate. When the prior information is correct, there is no doubt that OE will have lower posterior uncertainty since OE can leverage an extra source of information to more efficiently derive its estimate. However, this efficiency comes at a potential cost when the prior is incorrect. For instance, we showed that when OE uses an incorrect prior mean, then the estimate is guaranteed to be biased. Estimates from the SVD method in the principal component basis, on the other hand, are insensitive to incorrect information coming from the prior. The choice between SVD and OE then mostly comes down to how well one understands the prior distribution of the state of interest."

My view:

Bayesian and non-Bayesian methods just answer two different questions. Bayesian methods (with realistic prior) tell us what the most probable state of the atmosphere has been, while non-Bayesian methods tell us what the most plausible interpretation of the measurement is. None of these is superior; different questions demand different answers. But a Bayesian formalism with uniformative prior or ad hoc prior, I would say, are a variant of constrained maximum likelihood retrievals in disguise. I admit that Bayes himself endorsed uninformative prior, and also Gauss did it, but this has been heavily criticized by Pearson and Fisher, and we cannot be sure what Bayes himself thought about this issue because he did not publish his work while still alive.

Comment:

While it is true that the basis functions may not need to be constrained if truncated at whole degrees of freedom, and there may be no biases in the mapped space, the translation of the basis functions into a profile can result in biases and these should be quantified.

Reply:

The translation of the basis functions into a profile can indeed result in biases. We did show analytic expression for the bias of OE and SVD in the original profile space in the paper. They are in equation (30) and (33). For your convenience, we include them in this response. Suppose that OE and SVD both uses a wrong prior mean xb, which is different from the true prior mean xa, then the expected bias for OE is BiasOE = ... And the SVD bias is BiasSV D = ... Note that when OE uses the correct prior mean (xb = xa), then BiasOE = 0. Also, when (KSKt-1 is invertible, then BiasSV D = 0 regardless of the choice of xb. Detailed error analysis with simulations was done in the z-basis because the full GHG profile (x-basis) is often not needed for use in GHG flux modeling. The retrieved parameters obtained from the z basis retrievals can often be directly mapped to a GHG column mean and other higher order components. It should be noted that this is the basis of line-fitting methods even if they don't explicitly use the SVD method, since they derive XGHG strictly from the measurement, whose information is contained within the SVD basis. XGHG is ingested or assimilated into GHG flux models today. Higher order components like the vertical dipole moment can also be similarly ingested based on their information content as has been described in Joiner and DaSilva, "Efficient methods to assimilate remotely sensed data based on information content" (1998). A reference to Joiner and Da Silva (1998) has been added. A sentence has been added to the end of the last paragraph in section 2.1 "Joiner and Da Silva (1998) describe a method that can ingest such components into an assimilation model based on their information content."

My view:

I still think that, in the general case, truncated SVD can cause a bias, even in

the z-space. See my own comments below.

Comment:
The biases introduced by this approach should estimated by calculating the linear estimate for different true states, e.g. Tukiainen (2006) Fig. 3 shows the difference between AirCore and smoothed AirCore for methane and a similar SVD approach. The column difference between AirCore and smoothed AirCore (or some other set of trues) would give the bias and error resulting from the SVD mapping using Eq. 34 from this paper.

Reply:
The intent of the SVD approach described is for functional retrievals that can provide inputs into GHG flux models either in the absence of prior GHG profile information or when such information is of unknown quality or potentially biased. As stated above, these do not require a retrieval of a full vertical profile. Nevertheless, we have shown the expression for calculating the error in the GHG profile. Please see Eqn (1) and (2) from the response above.

My view:
Here the authors miss the point the different states result in different K-matrices. I think this is why the reviewer wants to see linear estimates for different true states. On the other hand, I have no idea how state-dependent the K matrix is in the given application. Possibly an adequate caveat could save the paper without much additional investigation.

Comment:
The authors should clarify how many basis functions are selected. If there are 1.6 degrees of freedom, are 2 basis functions used? If 2, won't the retrieval need some constraint? If 1, won't that throw away information? Kulawik et al. (2017) used 1.6 degrees of freedom from GOSAT to get 2 parameters each with about 0.8 degrees of freedom (so some a priori component in the retrieved values). Would this approach be able to get any vertical information with 1.6 degrees of freedom?

Reply:
The calculation of degrees of freedom depends on the state of prior knowledge of the true state. In the SVD framework as we have used it, with prior information being absent, the retrieval process works the same whether the prior GHG profile is well known or poorly known. The key metric is the posterior uncertainty of the retrieved quantities. One can compare the posterior uncertainty against that of the prior knowledge of the system. With retrievals in the principal component basis, the retrieved parameters are orthogonal to each other. The practical implication of this is that the mean and variance of a retrieved parameter is independent of the inclusion or exclusion of higher order parameters. This is illustrated in Figures 10 and 11. If a higher order parameter is retrieved despite insufficient information, that parameter will have a posterior

uncertainty that is too large to be useful. If 2 degrees of freedom are retrieved when the Bayesian framework projects 1.6 degrees of freedom, the second order component will have an uncertainty that is slightly too large to be useful on its own. As mentioned previously, it will not affect the retrieval otherwise.

My view:
A naive question: Why not considering only a fraction of the critical component, e.g., when reconstructing the profile in the x-space, on might consider the first singular vector in full and the second only with reduced weight? But such a modification of the method - if possible at all - would be beyond the scope of the paper. Given the application of the data intended and discussed in the paper, the authors' argument is compelling.

Comment:
I do not follow Figure 4. What are the units on the y-axis? Are the authors aware that OCO-2 has better precision and more degrees of freedom than GOSAT? This figure suggests the opposite.

Reply: We are aware that OCO-2 has better prevision and more degrees of freedom than GOSAT. The purpose of Figure 4 was to illustrate the effect of spectral resolution, particularly on the higher order components. The overall precision depends on several factors including the light gathering capacity, detector sensitivity, integration time, etc. We have replaced the words GOSAT and OCO-2 with regions indicating spectral resolutions of satellite instruments. Here is the new Figure 4 with caption Figure 1: (Figure 4 in manuscript) Retrieval uncertainty versus instrument spectral linewidth for the first two CO2 principal components (PCs) : While the column XCO2 is largely unaffected by the spectral resolution, the precision of the CO2 vertical dipole moment degrades strongly with poorer resolution. We assume a CO2 instrument model, but with some instrument line broadening. The x-axis denotes the full-width at half maximum of the triangular instrument line shape used to broaden the CO2 absorption. We assume photon shot noise with a SNR of 1000 for points with no CO2 absorption. The spectral resolutions of TCCON, satellite GHG sensing spectrometers and the CO2 Sounder instrument are indicated, though the calculations done in this work apply only to the CO2 Sounder instrument.

My view:
The purpose of this part of the paper is a methodical study, not a comparison between two existing instruments. I consider the action taken by the authors in reply to the revieM as adequate.

Comments:
I look forward to reviewing this paper when the authors improve the OE results and more carefully characterize the bias and errors compared to the current OE method.

My view:
The authors and the reviewer seem not to agree what the scope of the paper shall be. If the paper is understood as a methodical study and not as an evaluation of existing instrument concepts, the criticism by the reviewer appears a bit too harsh to me.

Other changes by the authors (I do not comment on all of them):

We have replaced the use of the word "bias-free" in relation to the retrieved SVD components with a more accurate description, i.e that the components do not incur bias from the regularization process or 7the use of an uninformative prior.

My view:
This does not fully solve the problem; this is because $K$ can depend on $x$. A different $x_u$ can thus still lead to a different result. See also my own comment below. What is needed here is either an assessment that the dependence of K on x is weak enough to be neglected or a caveat that this type of effects can exist but has not been assessed.

My own comments (pagination refers to amt-2018-14-manuscript-version3.pdf):

p2 l16 Hansen (1990) -¿ (Hansen, 1990)

p2 l17 The statement that the retrieved principal components are unbiased or bias-free may be true in the given context but not in general. Assume a thermal emission instrument where the signal depends on the temperature of the emitting layer. The SVD method will change the profile shape by removing fine structure. With this a certain amount of gas may be shifted into another altitude where the temperature is different. Thus, the same amount of gas may generate a different amount of radiance, and in turn, the retrieved total amount of gas depends on the assumed profile shape. This counter-example may sound contrived but at least it disproves the general validity of the statement made. This mechanism may become effective only within an iterative context. But even without considering an iterative process, $K$ depends on $x_u$, even if the uninformative prior is realized by setting $S_a^{-1}$ zero.

p2 l30: The term "uninformative prior" plays an essential role in this paper, thus it needs to be clearly defined when first used.

p4 l12 Not clear why least squares fit is needed if the retrieval problem is fully determined. A direct solution by matrix inversion would be possible. Least squares fit is only needed if the problem is OVER-determined. It is clear what the authors mean but the wording is a bit sloppy here and may direct the reader into a wrong direction.

p4 21-27 These statements are certainly correct but usually this problem is solved by distinguishing between "variables" and "parameters" of the retrieval problem; those input values of the forward model which are kept constant during the retrieval can be called parameters and those which are part of the x-vector are the variables. I think this would comply with traditional language in the case of a function F with multi-dimensional input. The term 'a priori' then could be reserved for the latter ones. This terminology would make this paragraph obsolete but would be in conflict with the terminology in the remainder of the paper (e.g. Fig 1 where what I call 'variables' is called 'parameters'. Probably it is the best to leave the terminology as it is, because it is, at least, self-consistent.

p5 l13 (I know that I am exaggeratedly fussy with such issues!) I do not quite agree that one really can gradually move back and forth between Bayesian and non-Bayesian methods. The reason is this: The solution of a Bayesian retrieval can be interpreted in terms of probability. The solution maximizes the a posteriori probability. A method which does not use the prior information will not render a solution which can be construed as the maximum of a probability distribution but should be understood in the sense of likelihood (Fisher, 1922). The move from 'conceivable as probability' to 'not conceivable as probability' is discontinuous, even if formally and result-wise the move is continuous. I think the problem can easily be remedied by writing "...from fully-Bayesian-like formalism to ...". Then it is clear that there is no transition between the concepts behind the formulas and no claim is then made that the Bayesian formalism really represents the concept of maximization of the a posteriori probability.

p5 l11: (I am still fussy...) Is there really a "choice of the prior covariance matrix". I know that this terminology is often used as internal slang of the retrieval community; but is, in a Bayesian sense, the a priori covariance matrix really something one can 'choose'? I suggest a slightly modified language where, whenever an ad hoc choice of the a priori is used, the term 'a priori ASSUMPTION' is used, and a ... matrix IN PLACE of the a priori covariance matrix.

Eqs 16-22 and related text: What you actually show (because you set $_Sa^{-1}$ to zero) that the SVD retrieval is equivalent with a simple Gaussian weighted least squares retrieval admittedly, Gauss also used a non-informative prior to give this approach a probabilistic interpretation. This comes down to a maximum likelihood retrieval (Fisher, 1922). I find it misleading to claim to have shown the equivalence with an OE retrieval, because the main characteristic of the latter as understood today is that it does use INFORMATIVE prior.

Sect 3.5. See my comment above: You formally prove that the SVD method is bias-free in a sense that the SVD concept does not introduce a bias and that the prior remains ineffective. This seems to be in conflict with the bias-causing mechanism I propose above. The reason for this conflict is this. The formal proof uses linear algebra. My proposed mechanism is nonlinear because it considers effects which are caused by the profile-dependence of K. The SVD

tyoically removes fine structure of the profile. Imagine the the atmosphere is particularly hot where a peak in the profile is located. Let SVD remove this peak, in a way that the total column is unchanged. With the same amount of molecules, the forward model now produces less radiance, and the retrieval will put in additional molecules to compensate for this. In other settings than thermal emission, e.g. reflected solar radiance, the effect may be less pronounced but absorption cross-sections are also temperature dependent; and there may be further mechanisms which might cause a profile-dependence of $K$. The truncated SVD approach changes the profile (in the $x$-space to which $z$ has to be transformed back if F is to be evaluated in the second iteration) this approach may cause a bias. And as said above, even in a non-iterative context, $K$ depends on $x_u$, even if $Sa^{-1}$ is set to zero. Thus, via $K$, the prior is not as uninformative as it may appear. It was very audacious to claim that the method is always bias-free, and even the wording in the revised version still appears too strong to me. I suggest to add a caveat to the text, like "within linear theory, i.e. without consideration of the profile-dependence of $K$" or something similar.

Sect 4.4.: The concepts of resolution and sampling are often confused. I thus appreciate that here it is between both these concepts. However, I have two suggestions to make this paragraph even clearer.
1. p19 l20: Here a 'smaller' resolution is mentioned. Language is (in general, not only here) confusing because if a small number is associated with the resolution, the resolution is good and if a large number is mentioned, the resolution is poor. When a 'small' resolution is mentioned, it is not clear if the resolution is good (low number) or if the resolution is poor (high number, low resolving power); a terminology using good vs. poor resolution would less unambiguous.
2. p19 l22: I suggest to avoid to combine the terms 'resolution' and 'sampling', because these concepts are too often confused. Thus I suggest to replace 'high sampling resolution' with 'dense sampling'.

Sect. 6.4. The most relevant difference between the Bayesian method (with an a priori which correctly describes the background statistics) and other methods is that it does, contrary to all other methods, renders the most probable solution. Methods which do not, in one way or another, invoke the Bayes theorem don't do this.

By the way, SVD is not the only rationally founded way to reduce the effective dimension of the retrieval. The common goal is to make the retrieval stable without becoming explicitly dependent on prior information. Which of these methods is the most adequate depends on the intended application of the data, and I appreciate that the authors raise this issue. A particular disadvantage of OE (with informative prior) is that the averaging kernels depend on the atmospheric state and thus on time, which makes time series hard to understand and interprete. Similarly, the singular vectors of an SVD method depend on the atmospheric state. Von Clarmann et al. (2015) suggest a different method where the retrieval basis is time-independent (but this may be a bit off-topic

here).

SUMMARY:

In summary, it seems to me that all issues can be remedied by purely redactional actions without much further scientific inverstigation. Thus I recommend publication after minor revision.

---

## Author Response (AR2)

Here are the final responses to the comments of Reviewer 3 with the reviewer comments in **bold** and our response below that

1. **As stated below in more detail, this is not generally true but only if the signal measured is independent of any other altitude-dependent atmospheric state variable. Care should be taken to avoid any generalizing statement on bias-freeness but to limit the statements to particular results of the investigations made.**

   We believe the crux of the issue here lies in our assumption that the Jacobian matrix $\mathbf{K}$ is *constant*, at which point the SVD methodology is unbiased. As pointed out in the reviewer comments, this does not necessarily hold when Jacobian matrix $\mathbf{K}$ is not constant across the domain. We've updated the draft to emphasize that we are assuming a constant $\mathbf{K}$, and that the properties derived therefrom are applicable only when the assumptions are valid (or mostly valid).

   We agree that when the forward model is not linear, then a bias would be possible under the SVD retrieval. However, to be fair, we note that a perfect OE retrieval (i.e., having the correct forward model and priors) would also be susceptible to bias under a non-linear forward model. This is readily apparent when we consider that the OE estimate is a maximum a posteriori estimate, but an estimate is only unbiased (that is, $E(\hat{\mathbf{x}} - \mathbf{x}) = \mathbf{0}$) when it is equal to the posterior mean. When the forward model is linear, then OE's maximum a posteriori estimate is equal to the posterior mean, but this equality generally does not hold when the forward model is non-linear. We've added a section below to the end of Section 3.5,

   "As a caveat, we note that the bias derivation above assumes that the forward model is linear for both the OE and SVD retrieval (as is the case for GHG retrievals discussed here, see section 2.3), and therefore the bias equations for OE and SVD (Eqn 30 and 31) should only hold when the assumption is true or mostly true. For the more general non-linear forward model, both the OE and SVD retrievals might be biased, but a thorough exploration of this non-linear case beyond the scope of this paper. "

   Also, we will add that a popular approach for non-linear forward model is to assume that the model is linearizable in some region and assume that the linear analysis extend to this region. This is essentially how Clive Rodgers derived the retrieval uncertainty that is now the de facto standard for OE uncertainty (e.g., OCO-2 XCO2 retrievals). In the

same way, we expect that our linear analysis of SVD should apply whenever the linearization assumption is valid.

2. **There are many retrieval methods which aim at minimizing the bias of the retrieval by avoiding informative prior and which keep the result stable basically by reducing the effective dimension of the result vector. Tukiainen (2016) is one of these, von Clarmann et al. (AMT 8, 2749-2757, 2015) would be another one, and I am sure there are a lot more. Comparison with each of these methods would be interesting but a line must be drawn somewhere and the decision which of these comparisons should be included in the paper should be left to the authors. The authors should not be forced to discuss any specific one of these papers unless a compelling reason is given that paper x is more important than paper y.**

   We thank the reviewer for these comments.

3. **Again I agree with the authors. The purpose of the related section is validation, not competition. A common practice of validation is comparison with something well understood and easily traceable. The choice of a simplified OE approach appears to be rational and justifiable to me.**

   Again, we thank the reviewer for these comments.

4. **While I agree with the argument of the authors in its heart, I have some reservations with the use of the term "OE" in the context of ad-hoc priors or uninformative priors. See my own comments below. Bayesian and non-Bayesian methods just answer two different questions. Bayesian methods (with realistic prior) tell us what the most probable state of the atmosphere has been, while non-Bayesian methods tell us what the most plausible interpretation of the measurement is. None of these is superior; different questions demand different answers. But a Bayesian formalism with uninformative prior or ad hoc prior, I would say, are a variant of constrained maximum likelihood retrievals in disguise. I admit that Bayes himself endorsed uninformative prior, and also Gauss did it, but this has been heavily criticized by Pearson and Fisher, and we cannot be sure what Bayes himself thought about this issue because he did not publish his work while still alive.**

We agree with the reviewer that a Bayesian formalism with uninformative prior or ad hoc prior, is not different from constrained maximum likelihood retrievals. We understand the reservation on using "OE" with an uninformative prior or "ad hoc" prior, but in keeping with common convention we have chosen to describe such 'OE' methods (including ad-hoc priors) as Bayesian, since that is how it is described in Rodgers (2000), and the several papers that cite it.

We note that in practice, specifying a prior on a multivariate state $\mathbf{x}$ is typically difficult, and instrument teams that are using OE methods effectively use "ad hoc" priors in that their prior is a combination of physical fidelity, computational feasibility, and expediency. This approach is reflected in most implementations of OE retrievals. For instance, the OCO-2 retrieval uses a state vector that includes carbon dioxide, aerosol properties, and surface properties. The prior covariance is assumed to be diagonal for all non-$CO_2$ elements. For the $CO_2$ elements, the prior covariance has off-diagonal elements "estimated based on the Laboratoire de Meteorologie Dynamique general circulation model, but the correlation coefficients were reduced arbitrarily to ensure numerical stability in taking its inverse" (Boesch et al., 2015). Furthermore, the diagonal elements of the $CO_2$ prior covariance are "unrealistically large for most of the world (all relatively clean-air sites), [they are] intended to be a minimal constraint on the retrieved XCO2."

Another example of the compromise between expediency and physical fidelity in designing the prior distribution of the state can be seen in Irion et al.'s (2017) OE retrieval of AIRS temperature and water vapor. There, the state vector consists of surface temperature, atmospheric temperature, water vapor, $CO_2$, $O_3$, and cloud properties. The prior covariance matrix is block diagonal with no covariance between any of the respective constituents (e.g., between temperature and water vapor). Temperature, however, is assumed to have an exponential covariance structure (also called Markov process covariance, see Rodgers, 2000) along the vertical direction. The same structure is assumed for water vapor, $CO_2$, and $O_3$, albeit with different length scales whose values are "guided by previous experience with AIRS and TES retrievals" (Irion et al., 2017).

These examples show that in practice, OE retrieval tend to be rather "loose" in their choice of prior, which usually is a mixture of two traditions. In the first, the prior is viewed in the Twomey-Tikhonov approach, where the prior is considered as a "regularizer" or penalty to ensure good convergence properties. The second approach is the

Bayesian approach where the prior should capture the real geophysical dependence between the components therein. That is, it should be as physically realistic as possible. Take the OCO-2 prior covariance matrix in the operational code, for instance. In this matrix, the components of aerosol and surface properties are assumed to be independent of one another, an ad-hoc assumption that is not reflective of reality (Twomey Tikhonov approach). The $CO_2$ components, on the other hand, have a correlation structure that is based on model data (Bayesian approach). Furthermore, the CO2 components (the $20 \times 20$ subblock of the entire covariance matrix) is then inflated by a factor of about 100 to "be a minimum constraint on the retrieved XCO2" (shades of the uninformative prior approach).

5. **I still think that, in the general case, truncated SVD can cause a bias, even in the z-space. See my own comments below. Here the authors miss the point the different states result in different K-matrices. I think this is why the reviewer wants to see linear estimates for different true states. On the other hand, I have no idea how state-dependent the K matrix is in the given application. Possibly an adequate caveat could save the paper without much additional investigation.**

   We agree that when the forward model is non-linear, then there most likely would be a bias. However, in the specific case of GHG remote sensing that we discuss in the paper, the forward model is linear (that is, $\mathbf{F}(\mathbf{x}) = \mathbf{Kx}$). We added a caveat of this for emphasis in the first paragraph of Section 2.3. We would also make a note that typical variances in column GHG mixing ratios (especially $CO_2$ and $CH_4$) mostly do not vary by more than a few percent.

6. **A naive question: Why not considering only a fraction of the critical component, e.g., when reconstructing the profile in the x-space, on might consider the first singular vector in full and the second only with reduced weight? But such a modification of the method - if possible at all - would be beyond the scope of the paper. Given the application of the data intended and discussed in the paper, the authors' argument is compelling.**

   We agree with the reviewer that there are other possibilities of using the SVD components in reconstructing the profile in the $\mathbf{x}$-basis, some of which are smoother than the hard truncation that we use.

7. **The purpose of this part of the paper is a methodical study,**

not a comparison between two existing instruments. I consider the action taken by the authors in reply to the reviewer as adequate.

We thank the reviewer for these comments.

8. **This does not fully solve the problem; this is because K can depend on x. A different $\mathbf{x}_u$ can thus still lead to a different result. See also my own comment below. What is needed here is either an assessment that the dependence of K on x is weak enough to be neglected or a caveat that this type of effects can exist but has not been assessed.**

We agree that we should add caveats about the assumption of constant Jacobian in the paper. Please see our responses for comments 1 and 5.

9. **p2 l16 Hansen (1990) -¿ (Hansen, 1990)**

We have fixed the typo.

10. **p2 l17 The statement that the retrieved principal components are unbiased or bias-free may be true in the given context but not in general. Assume a thermal emission instrument where the signal depends on the temperature of the emitting layer. The SVD method will change the profile shape by removing fine structure. With this a certain amount of gas may be shifted into another altitude where the temperature is different. Thus, the same amount of gas may generate a different amount of radiance, and in turn, the retrieved total amount of gas depends on the assumed profile shape. This counter-example may sound contrived but at least it disproves the general validity of the statement made. This mechanism may become effective only within an iterative context. But even without considering an iterative process, K depends on $\mathbf{x}_u$, even if the uninformative prior is realized by setting $\mathbf{S}_a^{-1}$ zero.**

We've added caveats to the paper about this linear assumption. Please see the responses to comments 1 and 5. We have changed the sentence on p2 l17 to remove the phrase "opening the possibility of an unbiased retrieval".

With the above example, there is a larger point with regards to GHG molecules having different cross-sections depending on where they are in the atmosphere. Thus, GHG molecules moving to a different part of the atmosphere will change the retrieved column 'mean'. It should

be noted that what is commonly referred to as the column mean in the retrieval community (including this work) should not be construed as a true column mean, *i.e.* one that has a flat averaging kernel and is thus insensitive to vertical transport of GHG molecules. We have added the following sentences to section 4.3 (Averging Kernels)

"It should be noted that what is commonly referred to as the column mean in the retrieval community (including this work) should not be construed as a true column mean, *i.e.* one that has a flat averaging kernel and is thus insensitive to vertical transport of GHG molecules. As one can see in Figure 3 (right), the 'column mean' averaging kernel has some vertical dependence. "

11. **p2 l30: The term "uninformative prior" plays an essential role in this paper, thus it needs to be clearly defined when first used.**

    We have added a paragraph under section 2.1 (Regularization of the retrieval problem and vertical information)

    "An uninformative prior is one that fills in information necessary for a retrieval (here a GHG profile) but it tries to be as vague as possible. In this paper, our uninformative prior makes use of the principle of indifference, which assigns equal probability to all possibilities."

    We've also added a note about this prior in the second paragraph of Section 3.3

    "We note that the uninformative prior distribution $N(\mathbf{x}_a, \mathbf{S}_a \to \infty)$ is technically an improper prior in that it is not a well-defined probability distribution. However, it does yield a well-defined Bayesian posterior distribution."

12. **p4 l12 Not clear why least squares fit is needed if the retrieval problem is fully determined. A direct solution by matrix inversion would be possible. Least squares t is only needed if the problem is OVER-determined. It is clear what the authors mean but the wording is a bit sloppy here and may direct the reader into a wrong direction.**

    We agree with the reviewer. We have changed the paragraph to provide the right context:

    *"Even when the number of measurement samples far exceed the number of retrieved parameters (as with column GHG absorption measurement spectra), retrieval problems may or may not be fully determined*

*depending on the information content of the samples with respect to the retrieved parameters. In situations where the retrieval problem is fully determined, one can obtain a unique solution of the parameters of interest. When the problem is over-determined, one can perform a least-squares fit to solve for the parameters of interest."*

13. **p4 21-27 These statements are certainly correct but usually this problem is solved by distinguishing between "variables" and "parameters" of the retrieval problem; those input values of the forward model which are kept constant during the retrieval can be called parameters and those which are part of the x-vector are the variables. I think this would comply with traditional language in the case of a function F with multi-dimensional input. The term 'a priori' then could be reserved for the latter ones. This terminology would make this paragraph obsolete but would be in conflict with the terminology in the remainder of the paper (e.g. Fig 1 where what I call 'variables' is called 'parameters'. Probably it is the best to leave the terminology as it is, because it is, at least, self-consistent.**

We agree with the reviewer and have thus left it as is.

14. **p5 l13 (I know that I am exaggeratedly fussy with such issues!) I do not quite agree that one really can gradually move back and forth between Bayesian and non-Bayesian methods. The reason is this: The solution of a Bayesian retrieval can be interpreted in terms of probability. The solution maximizes the a posteriori probability. A method which does not use the prior information will not render a solution which can be construed as the maximum of a probability distribution but should be understood in the sense of likelihood (Fisher, 1922). The move from 'conceivable as probability' to 'not conceivable as probability' is discontinuous, even if formally and result-wise the move is continuous. I think the problem can easily be remedied by writing "...from fully-Bayesian-like formalism to ...". Then it is clear that there is no transition between the concepts behind the formulas and no claim is then made that the Bayesian formalism really represents the concept of maximization of the a posteriori probability.**

We are of a different opinion in that we consider the difference between Bayesian and non-Bayesian to come down to a matter of how one is

willing to introduce beliefs in an analysis (see also the response to the next Comment). The prior that is used in Bayesian statistics is often interpreted as a statement of subjective belief, and we subscribe to this opinion. We have shown in this paper that the improper uninformative prior $\mathbf{S_a} \to \infty$ is mathematically equivalent to maximum likelihood in that they would produce the exact same estimates. While the distribution $N(\mathbf{x_a}, \mathbf{S_a} \to \infty)$ isn't a well-defined probability distribution, its use as a prior can result in a well-defined posterior probability distribution, as derived in Section 3, and it still makes sense to talk about maximization of the posteriori probability for the case where the prior has the form $N(\mathbf{x_a}, \mathbf{S_a} \to \infty)$.

So perhaps it is better to say that in this paper we are saying that in Bayesian statistics (of which OE retrieval is an example) one can move gradually between informative priors and the uninformative priors. As a Bayesian, there is no issue with using an uninformative prior and indeed Bayesian statisticians have been using them for a long time. We also note that some instrument teams in practice are also flirting with the uninformative prior by inflating certain sub-block of their prior covariance matrix "in order to impose minimal constraint on the retrieval" (see description of OCO-2 prior in the response to Comment 4). It just happens that our particular choice of uninformative prior here is mathematically equivalent to maximum likelihood.

15. **p5 l11: (I am still fussy...) Is there really a "choice of the prior covariance matrix". I know that this terminology is often used as internal slang of the retrieval community; but is, in a Bayesian sense, the a priori covariance matrix really something one can 'choose'? I suggest a slightly modified language where, whenever an ad hoc choice of the a priori is used, the term 'a priori ASSUMPTION' is used, and a ... matrix IN PLACE of the a priori covariance matrix.**

The question "is there really a choice of the prior covariance matrix?" brings to mind the famous joke that if one poses the same question using the same data to three Bayesian statisticians, then one would get three different answers. Indeed, if we give the same radiance vector and forward model to three different retrieval teams, then we very well might get three different retrievals depending on what the teams come up with for the prior mean and prior covariance.

We are of the opinion that the prior used in Bayesian methods is really expressing a subjective "belief" about what one might think about

the distribution of the state **x**. I don't think there is any retrieval team in this world who can definitely say that they know the 'true' prior distribution of their state **x**. The best they could do is making a statement about what they think it *might* be. Often, the priors that they come up in practice is a mixture of physical fidelity and expediency (see the response to Comment 4).

Nevertheless, the suggested change in wording is a good one and one which we think is consistent with how the prior is viewed *in practice*. Therefore we have made those changes.

16. **Eqs 16-22 and related text: What you actually show (because you set $S_a^{-1}$ to zero) that the SVD retrieval is equivalent with a simple Gaussian weighted least squares retrieval admittedly, Gauss also used a non-informative prior to give this approach a probabilistic interpretation. This comes down to a maximum likelihood retrieval (Fisher, 1922). I find it misleading to claim to have shown the equivalence with an OE retrieval, because the main characteristic of the latter as understood today is that it does use INFORMATIVE prior.**

The point above seems to be that OE retrieval has to use an *informative* prior, and we take a slightly different view. The crux of the OE retrieval is essentially given a prior $P(\mathbf{x})$ and the radiance vector **y**, then the OE retrieval would attempt to quantify the posterior distribution $P(\mathbf{x}|\mathbf{y})$. The mathematics of OE impose no restriction on what the prior $P(\mathbf{x})$ has to be. And if we take the position that the prior represents beliefs, then one could believe whatever one wants to believe (obviously subject to some scientific justification). In our perusal of Rodgers (2000), we did not find a discussion of whether OE has to use an informative prior. However, we note that OE retrieval is often considered as a Bayesian methodology, and in the Bayesian literature it is perfectly *de rigueur* to use an uninformative prior.

We acknowledge that the existing publication on OE retrieval invariably use an informative prior. However, we submit that 1) OE retrieval already has the framework to admit non-informative priors (and indeed, in practice some instrument teams are already flirting with them by inflating certain sub-components of the prior covariance- see the discussion of OCO-2 operational prior covariance above) and 2) the OE theoretical framework would be richer if we do not force the prior to be informative.

We also note here that when it comes to prior $\mathbf{S_a}$, the word "informative" typically means "finite" and the dividing line between informative and uninformative can be a bit nebulous in practice. This is essentially what the OCO-2 team had in mind when they inflated the $CO_2$ sub-block of their matrix $\mathbf{S_a}$ to minimize impact on retrievals. Playing upon this grey area between informative and uninformative prior further, we could make an informative prior that is indistinguishable (in the computer numerical sense) from an uninformative prior by simply inflating it. That is, if we let $\mathbf{S_a} = \mathbf{10^{100}} \times \mathbf{I}$ where $\mathbf{I}$ is the identity matrix, then in practice the OE retrieval with this informative prior would return the same estimates as one using the uninformative prior.

17. **Sect 3.5. See my comment above: You formally prove that the SVD method is bias-free in a sense that the SVD concept does not introduce a bias and that the prior remains ineffective. This seems to be in conflict with the bias-causing mechanism I propose above. The reason for this conflict is this. The formal proof uses linear algebra. My proposed mechanism is nonlinear because it considers effects which are caused by the profile-dependence of K. The SVD typically removes fine structure of the profile. Imagine the the atmosphere is particularly hot where a peak in the profile is located. Let SVD remove this peak, in a way that the total column is unchanged. With the same amount of molecules, the forward model now produces less radiance, and the retrieval will put in additional molecules to compensate for this. In other settings than thermal emission, e.g. reflected solar radiance, the effect may be less pronounced but absorption cross-sections are also temperature dependent; and there may be further mechanisms which might cause a profile-dependence of K. The truncated SVD approach changes the profile (in the x-space to which z has to be transformed back if F is to be evaluated in the second iteration) this approach may cause a bias. And as said above, even in a non-iterative context, K depends on $\mathbf{x}_u$, even if $\mathbf{S}_a^{-1}$ is set to zero. Thus, via K, the prior is not as uninformative as it may appear. It was very audacious to claim that the method is always bias-free, and even the wording in the revised version still appears too strong to me. I suggest to add a caveat to the text, like within linear theory, i.e. without consideration of the profile-dependence of K" or something similar.**

The reviewer raises an important concern. we have discussed the profile dependence of $\mathbf{K}$, and how it ultimately does not lead to bias (see response to 11 above). Our example in section 5 where we have used an uninformative prior that differs sharply from the actual profile further illustrates the point. We have also previously addressed the concern about the altitude dependence of the absorption cross-section (see response to 10 above).

18. **Sect 4.4.: The concepts of resolution and sampling are often confused. I thus appreciate that here it is between both these concepts. However, I have two suggestions to make this paragraph even clearer. 1. p19 l20: Here a 'smaller' resolution is mentioned. Language is (in general, not only here) confusing because if a small number is associated with the resolution, the resolution is good and if a large number is mentioned, the resolution is poor. When a 'small' resolution is mentioned, it is not clear if the resolution is good (low number) or if the resolution is poor (high number, low resolving power); a terminology using good vs. poor resolution would less unambiguous. 2. p19 l22: I suggest to avoid to combine the terms 'resolution' and 'sampling', because these concepts are too often confused. Thus I suggest to replace 'high sampling resolution' with 'dense sampling'.**

    This is an excellent suggestion that makes the paper much clearer! We thank the reviewer for this suggestion and have incorporated it.

19. **Sect. 6.4. The most relevant difference between the Bayesian method (with an a priori which correctly describes the background statistics) and other methods is that it does, contrary to all other methods, renders the most probable solution. Methods which do not, in one way or another, invoke the Bayes theorem don't do this. By the way, SVD is not the only rationally founded way to reduce the effective dimension of the retrieval. The common goal is to make the retrieval stable without becoming explicitly dependent on prior information. Which of these methods is the most adequate depends on the intended application of the data, and I appreciate that the authors raise this issue. A particular disadvantage of OE (with informative prior) is that the averaging kernels depend on the atmospheric state and thus on time, which makes time series hard to understand and interpreted. Similarly, the singular vectors of an SVD method depend on the atmospheric**

**state. Von Clarmann et al. (2015) suggest a different method where the retrieval basis is time-independent (but this may be a bit o-topic here).**

We agree with the reviewer and thank the reviewer for the perspective. We have made no change relevant to this.

**Other minor changes for better readability and to fix typos**

1. In page 4, when discussing the relationship between the prior assumption and the retrieval result, we have changed the word "affects" to "can affect".

2. On page 7, line 21, we have corrected a typo in the subscript to $\mathbf{x}_u$

3. On page 9, we have changed an incorrect capitalization of $\mathbf{x}$

4. On page 12, line 24, we have changed the sides of an equation from $\mathbf{\Gamma}^T\mathbf{\Gamma} = \mathbf{D}$ to $\mathbf{D} = \mathbf{\Gamma}^T\mathbf{\Gamma}$

5. On page 14, line 24, we have changed "Negligible error from imperfect knowledge of..." to "Negligible error in the knowledge of .."

6. On page 15, first line, we have changed "Retrieval errors in the traditional OE method arise from ..." to "Retrieval errors in the traditional OE method can arise from ..."

7. On page 18, we have changed "... is the key limiting factor ..." to "... is the fundamental limiting factor ..."

8. On page 18, section 4.3, first paragraph discussing the terms resulting from the Singular Value Decomposition, we have added the following sentence: "This is to be expected since every wavelength sample is, in a sense, independently measuring the signal amplitude making it the most prominent term. "

9. On page 24, Fig. 8 - We have added a note to the figure caption for clarity : "(note - SVD and OE histograms are almost perfectly overlapped)"

10. We have added a sentence in Section 5.3 for more context: "We choose a sample profile from an atmospheric $CO_2$ profile measured from aircraft using an *in situ* instrument from an airborne campaign over California in 2016 (Abshire et al, 2018). "

11. We have updated the reference of Abshire et al, 2017 AMTD to Abshire et al, 2018 AMT

12. We have added a sentence at then end of section 6.4: " Finally, the SVD method can also be used to check the validity of an OE prior used for retrieval."

**References**

Boesch, H., Brown, L., Castano, R., Christi, M., Crisp, D., Eldering, A., Fisher, B., Frankenberg, C., Gunson, M., Granat, R., et al. (2015). Orbitng Carbon Observatory (OCO-2) Level 2 Full Physics Algorithm Theoretical Basis Document.

Irion, F., Kahn, B., Schreier, M., Fetzer, E., Fishbein, E., Fu, D., Kalmus, P., Wilson, R., Wong, S., and Yue, Q. (2017). Single-footprint retrievals of temperature, water vapor and cloud properties from AIRS. *Atmospheric Measurement Techniques, in review*, pages 1–35.

Rodgers, C. D. (2000). *Inverse methods for atmospheric sounding: theory and practice*, volume 2. World Scientific.

---

## Author Response (AR3)

We thank the Associate Editor for her suggestions on Figure 4 and have implemented them. We also updated a reference to the most recent submission on the CO2 Sounder instrument.